# MARS: OPTIMIZING DUAL-SYSTEM DEEP RESEARCH VIA MULTI-AGENT REINFORCEMENT LEARNING

## ABSTRACT

Large Reasoning Models (LRMs) often exhibit a tendency for overanalysis in simple tasks, where the models excessively utilize System 2-type, deliberate reasoning, leading to inefficient token generation. Furthermore, these models face challenges in adapting their reasoning capabilities to rapidly changing environments due to the static nature of their pre-training data. To address these issues, advancing Large Language Models (LLMs) for complex reasoning tasks requires innovative approaches that bridge intuitive and deliberate cognitive processes, akin to human cognition's dual-system dynamic. This paper introduces a **M**ulti-**A**gent System for Deep **ReS**earch (MARS) enabling seamless integration of System 1's fast, intuitive thinking with System 2's deliberate reasoning within LLMs. MARS strategically integrates multiple external tools—such as Google Search, Google Scholar, and Python Interpreter—to access up-to-date information and execute complex computations, while creating a specialized division of labor where System 1 efficiently processes and summarizes high-volume external information, providing distilled insights that expand System 2's reasoning context without overwhelming its capacity. Furthermore, we propose a multi-agent reinforcement learning framework extending Group Relative Policy Optimization to simultaneously optimize both systems with multi-turn tool interactions, bin-packing optimization, and sample balancing strategies that enhance collaborative efficiency. Extensive experiments demonstrate MARS achieves substantial improvements of 3.86% on the challenging Humanity's Last Exam (HLE) benchmark and an average gain of 8.9% across 7 knowledge-intensive tasks, validating the effectiveness of our dual-system paradigm for complex reasoning in dynamic information environments.

## 1 INTRODUCTION

Large Language Models (LLMs) have demonstrated remarkable capabilities across various tasks with their System 1's fast, intuitive thinking, yet they still struggle with complex reasoning tasks (Anthropic, 2024; Hurst et al., 2024; Yang et al., 2024; Mesnard et al., 2024; Team, 2025). Recent advances in this direction have led to the emergence of Large Reasoning Models (LRMs), which specifically excel at System 2-type, deliberate reasoning when confronted with challenging problems (Jaech et al., 2024; Guo et al., 2025; Team, 2024; Anthropic, 2025). However, when faced with simpler questions, LRMs may tend to overanalyze, producing an unnecessary amount of tokens Chen et al. (2024). In contrast to LLMs and LRMs, humans can effortlessly switch between these two modes of thinking.

Furthermore, the pursuit of developing LLMs capable of complex reasoning in rapidly changing environments remains a significant challenge. Since the knowledge these models possess is confined to the cut-off date of their training data, enabling them to acquire new information through web browsing—known as retrieval-augmented generation (RAG)—is emerging as a promising trend for enhancing reasoning capabilities (Chen et al., 2025; Li et al., 2025a; Jin et al., 2025; Li et al., 2025b). One of the most prominent examples is the proprietary system OpenAI (2025a); Google (2025b); xAI (2025); MoonshotAI (2025), the agents designed to synthesize large volumes of online information and accomplish multi-step research tasks effectively.

Inspired by the dual-process theory of human cognition (Evans & Stanovich, 2013; Frankish, 2010), we propose MARS, a multi-agent system for deep research to overcome previous limitations by innovatively integrating System 1 and System 2 thinking mode in a seamless manner, complemented

by external tools like *Google Search*, *Google Scholar*, and *Python Interpreter*. This interactive approach allows System 2 to focus on deliberate reasoning and planning, autonomously generating specific queries and computational tasks to engage with external resources. In turn, System 1 processes and summarizes these external tool outputs with its fast, intuitive thinking, distilling high-volume, potentially noisy information into concise insights for System 2's reasoning processes. The advantages of our proposal are manifest in two critical areas: first, through a specialized division of labor, our method considerably enhances the depth and breadth of available informational context in System 2. Specifically, System 1 efficiently filters and distills large volumes of retrieved information (such as multiple entire research papers or web pages) that would otherwise overwhelm System 2's reasoning capacity, thereby allowing the model to digest more comprehensive and up-to-date information without sacrificing reasoning quality. Second, this dual-system synergy creates an efficient collaborative framework where each component operates within its optimal domain, resulting in efficient yet robust problem-solving capabilities.

To implement our approach, we develop a data curation pipeline and collect a curated dataset from public datasets that emphasizes diversity across various difficulty levels and academic disciplines. This expansive dataset ensures that the model is well-prepared to tackle a broad spectrum of scenarios, ranging from straightforward tasks to intricate reasoning challenges, creating a solid foundation for subsequent training. To equip LLMs with these dual-system capabilities, we propose a multi-agent reinforcement learning framework that extends the Group Relative Policy Optimization (GRPO) (Shao et al., 2024) algorithm to optimize System 1 and System 2 simultaneously. In MARS, System 1 and System 2 function as collaborative agents implemented within the same underlying LLM but orchestrated through distinct prompts, enabling seamless interplay between intuitive and deliberate reasoning modes. Additionally, we propose several strategies to optimize our multi-agent training process. First, we incorporate multi-turn and multiple tools use during rollouts, allowing the model to dynamically refine its reasoning through iterative engagement with external tools. Second, we employ bin-packing algorithms (Coffman Jr et al., 1984) to efficiently organize variable-length retrieved content into optimally-sized chunks, significantly enhancing System 1's parallel processing efficiency. Third, to ensure balanced training dynamics between the two systems, we propose a two-phase approach that pre-computes each system's advantage signals and then implements a sample balancing strategy between systems, preventing one system from dominating the learning process. By integrating these strategies within our multi-agent RL framework, we enable both systems to adapt and refine their interactions, ultimately enhancing their collaborative efficiency and overall reasoning capabilities. Extensive experiments demonstrate that MARS achieves a substantial improvement of 3.86% on HLE benchmark and an average gain of 8.9% across 7 knowledge-intensive tasks, highlighting the effectiveness of our dual-system paradigm in complex reasoning scenarios.

In summary, our main contributions can be summarized as follows:

- We propose MARS, a novel dual-system framework that seamlessly integrates System 1's fast, intuitive thinking mode with System 2's deliberate reasoning capabilities, creating an efficient collaborative paradigm where each component operates within its optimal domain.

- We propose a multi-agent reinforcement learning framework to concurrently optimize System 1 and System 2, implementing multi-turn and multiple tools use for System 2, bin-packing optimization for System 1's efficient content processing, and pre-computation and sampling balance strategies to enhance collaborative efficiency and reasoning capabilities across diverse tasks.

- We contribute a data curation pipeline and a carefully curated training dataset spanning diverse difficulty levels and academic disciplines, with special emphasis on the challenging Humanity's Last Exam (HLE) benchmark, for which high-quality training data was previously scarce.

- We conduct comprehensive experiments on HLE and 7 knowledge-intensive tasks to evaluate the effectiveness of our MARS. Furthermore, we provide in-depth analysis of the multi-agent, multi-turn, multi-tool RL process and investigate how different external tools contribute to reasoning performance, particularly on the challenging HLE benchmark.

## 2 METHODOLOGY

In this section, we begin by providing a concise overview of the entire pipeline to help readers understand the comprehensive integration of System 1 and System 2 in deep research for tackling

complex questions. Then, we delve into the optimization strategies for end-to-end training of the two systems within a multi-agent reinforcement learning framework.

## 2.1 DUAL-SYSTEM COLLABORATIVE FRAMEWORK FOR DEEP RESEARCH

We design a collaborative framework that integrates System 1's intuitive processing capabilities with System 2's deliberate reasoning within a unified LLM. As illustrated in Figure 1, our framework establishes a synergistic workflow between these two systems to tackle complex questions through external tool utilization. Specifically, System 2 takes the lead in deliberate reasoning and strategically invokes external tools, while System 1 leverages its intuitive thinking to distill key information from these tool outputs. The communication between these

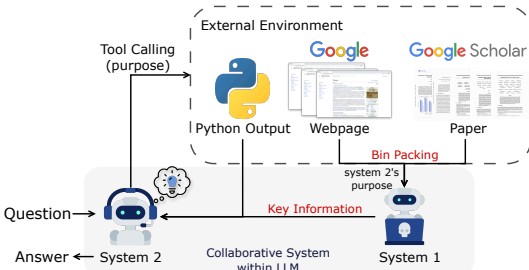

Figure 1: Overview of Dual-System Collaborative Framework in our MARS.

two systems is facilitated through the "purpose" of System 2's current tool invocation. This "System 2's purpose" serves as a crucial bridge, allowing System 1 to understand precisely what information to extract and summarize from potentially overwhelming external resources.

To formalize this collaborative framework, we represent these two systems activated within the same LLM through different prompts as $\pi_{\text{sys}_1}$ and $\pi_{\text{sys}_2}$. Given an initial question $q$ as the starting context $c_0$, we model the deep research process as a multi-turn interaction sequence. While System 2 maintains and reasons with the accumulated context $c_i$ (containing the question and information from previous turns), System 1 operates independently in each turn, focusing solely on processing the current tool outputs without requiring the full historical context.

In the $i$-th turn of interaction, the process unfolds as follows: (1) System 2 analyzes the current context $c_i$ and generates reasoning steps $s_i$, along with an optional tool request (which includes both the tool parameters $t_i$ and a specific purpose $p_i$):

$$s_i, (t_i, p_i) = \pi_{\text{sys}_2}(c_i) \tag{1}$$

where $t_i$ and $p_i$ can be empty if no tool is needed at this turn. (2) If $t_i$ is not empty, it is executed by the external environment, producing raw outputs $\{o_{t_i}^{(1)}, o_{t_i}^{(2)}, ..., o_{t_i}^{(n_{t_i})}\}$[1]. (3) If tool outputs are available, System 1 processes them to extract key information based on System 2's purpose $p_i$. To efficiently handle potentially large volumes of text (e.g., multiple web pages or research papers), we employ bin-packing algorithms (Coffman Jr et al., 1984) to organize the variable-length outputs into optimally-sized chunks that can be processed in parallel:

$$\tilde{o}_{t_i} = \pi_{\text{sys}_1} \left( \text{Bin-Packing} \left( o_{t_i}^{(1)}, o_{t_i}^{(2)}, ..., o_{t_i}^{(n_{t_i})} \right), p_i \right) \tag{2}$$

where $\tilde{o}_{t_i}$ represents the distilled information from all tool outputs. The specific bin-packing implementation details are discussed in Section 2.2.1. (4) The context is updated for the next turn by incorporating the reasoning, tool request, purpose, and the distilled information:

$$c_{i+1} = c_i \oplus \{s_i, t_i, p_i, \tilde{o}_{t_i}\} \tag{3}$$

where $\oplus$ denotes context concatenation. Note that for Python Interpreter outputs, $\tilde{o}_{t_i}$ is directly the tool output without System 1 processing, as these results are typically concise and structured. This process continues iteratively until System 2 determines to answer the original question. The overall generative process can be expressed as:

$$\mathcal{P}(\text{answer}|q) = \prod_{i=1}^{N} \left[ \underbrace{\pi_{\text{sys}_2}(s_i, t_i, p_i | c_i)}_{\text{System 2: Reasoning}} \cdot \underbrace{\pi_{\text{sys}_1}(\tilde{o}_{t_i} | \text{Bin-Packing}(o_{t_i}^{(1)}, o_{t_i}^{(2)}, ..., o_{t_i}^{(n_{t_i})}), p_i)}_{\text{System 1: Information Processing}} \right] \tag{4}$$

where $N$ is the total number of turns and the second term is omitted if no tool is called. This formulation clearly expresses how System 2 guides the overall reasoning process, while System 1 efficiently processes external information without maintaining the full context history.

---

[1] $n_{t_i}$ denotes the number of all outputs from request $t_i$. In our setting, Google Search returns up to 10 web pages per query, Google Scholar returns up to 5 papers, with multiple queries supported in a single tool request.

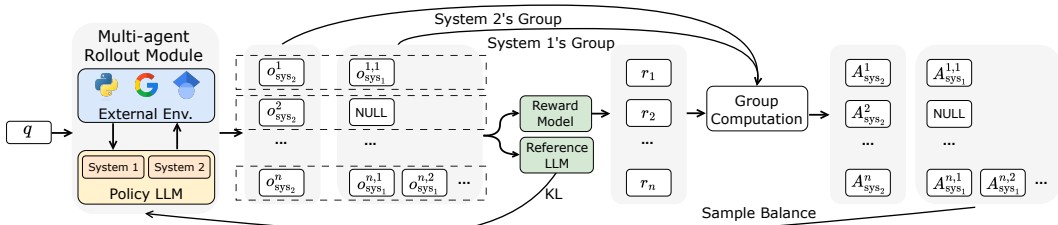

Figure 2: Demonstration of GRPO with multi-agent reinforcement learning in our MARS.

## 2.2 DUAL-SYSTEM OPTIMIZATION STRATEGIES

To maximize the effectiveness of our dual-system framework, we implement several key optimization strategies: (1) bin-packing algorithms to enhance System 1's parallel processing efficiency for variable-length retrieved content, and (2) advantage pre-computation and balanced sampling mechanism, preventing either system from dominating the learning process. Figure 2 illustrates our end-to-end RL training process, from multi-agent rollout to advantage computation and sample balancing.

### 2.2.1 EFFICIENT CONTENT PROCESSING WITH BIN-PACKING

During rollouts, System 2 follows standard token generation for deliberate reasoning and tool-use planning. However, the subsequent tool calls often return multiple outputs of variable length. Processing these large volumes of variable-length text presents a significant challenge for System 1 due to multiple generations. To address this, we employ an efficient bin-packing strategy based on the First Fit Decreasing (FFD) algorithm (Coffman Jr et al., 1984). This approach optimizes the organization of variable-length content into optimally-sized chunks to reduce the number of generations required by System 1.

Specifically, we begin by counting the number of tokens in each tool output $o_{t_i}^{(j)}$. If the token count exceeds System 1's maximum context length, the output is truncated and placed in an isolated bin. For the remaining outputs, we apply the FFD algorithm: all such outputs are first sorted in decreasing order of their lengths, then assigned to the first bin that can fit them, or to a new bin if no existing one suffices. We choose FFD over Best Fit Decreasing (BFD) due to its superior efficiency in practice. The implementation details can refer to Appendix D.2.

### 2.2.2 ADVANTAGE PRE-COMPUTATION AND BALANCED SAMPLING MECHANISM

In our MARS, following HLE's evaluation prompt (Phan et al., 2025), we employ LLMs as an evaluator to assess the predicted answer of each trajectory. Note that all System 1 and System 2 samples within the same trajectory share this trajectory-level reward to encourage both systems toward the same goal, rather than pursuing potentially conflicting individual objectives.

For each question $q$, we perform $G$ rollout trajectories and yield exactly $G$ System 2 samples and a variable number of System 1 samples, which depends on the number of tool calls per trajectory and chunks created after Bin-Packing. This unpredictable imbalance can lead to one system dominating the learning process, potentially undermining the collaborative dynamics essential to our multi-agent system. To address this, we pre-compute each system's advantage and then balance System 1's samples. Following GRPO (Shao et al., 2024), rewards are normalized within their corresponding groups to calculate advantage values:

$$A_{\text{sys}_2}^k = \frac{r_{\text{sys}_2}^k - \text{mean}(\mathbf{r}_{\text{sys}_2})}{\text{std}(\mathbf{r}_{\text{sys}_2})}, \qquad A_{\text{sys}_1}^{k,j} = \frac{r_{\text{sys}_1}^{k,j} - \text{mean}(\mathbf{r}_{\text{sys}_1})}{\text{std}(\mathbf{r}_{\text{sys}_1})}, \qquad (5)$$

where $\mathbf{r}_{\text{sys}_2} = \{r_{\text{sys}_2}^1, \ldots, r_{\text{sys}_2}^G\}$ represents all System 2 rewards for the current question, and $\mathbf{r}_{\text{sys}_1} = \{r_{\text{sys}_1}^{k,j} | k \in [1, G], j \in [1, n_k]\}$ represents all System 1 rewards, with $G$ being the group size, $n_k$ being the number of System 1 samples in trajectory $k$, and $r_{\text{sys}_1}^{k,j} = r_{\text{sys}_2}^k$.

After computing advantages for all samples, we implement a balanced sampling mechanism to align the number of System 1 samples with System 2. Specifically, if the total number of System 1 samples

$M = \sum_{k=1}^{G} n_k$ exceeds $G$, we randomly downsample to exactly $G$ samples; if $M < G$, we upsample through random duplication until reaching $G$ samples. This pre-computation-then-sampling approach offers two key benefits: First, it ensures that advantage information from all samples contributes to the computation before any sampling occurs, maximizing the utilization of available data. Second, it preserves the statistical integrity of the advantage distribution, as the normalization is performed across the complete set of samples rather than being distorted by the sampling process.

### 2.2.3 Multi-Agent Training Objective

With the balanced samples from both systems, we optimize System 1 and System 2 jointly using an extended GRPO framework, as shown in the Figure 2. The training samples for each system are distinctly different: For System 2, each sample consists of the full reasoning context $c_N = \{s_i, t_i, p_i, \tilde{o}_{t_i}\}_{i=1}^{N}$, where tokens from System 1's outputs $\tilde{o}_{t_i}$ are masked during loss computation. For System 1, each sample is a pair of bin-packed input and its corresponding output $(b, \tilde{o})$, where $b$ is a chunk created by the bin-packing algorithm and $\tilde{o}$ is System 1's response. The overall training objective combines the loss functions for both systems:

$$\mathcal{L}_{\text{total}} = \mathcal{L}_{\text{sys}_2} + \mathcal{L}_{\text{sys}_1} \tag{6}$$

For each system, we apply the GRPO objective (Shao et al., 2024):

$$\mathcal{L}_{\text{sys}_i} = \mathbb{E}_{(x,y)\sim\mathcal{D}_i} \left[ \mathcal{L}_{\text{policy}}(x, y, A_{\text{sys}_i}) + \lambda \mathcal{L}_{\text{KL}}(x, y) \right] \tag{7}$$

where $\mathcal{D}_i$ represents the balanced dataset for System $i$, $x$ and $y$ denote the input-output pairs, $\mathcal{L}_{\text{policy}}$ is the policy loss and $\mathcal{L}_{\text{KL}}$ is a KL regularization term. The detailed mathematical formulations are provided in Appendix D.4. This joint optimization approach allows both systems to improve simultaneously while maintaining their specialized roles in the collaborative framework.

## 3 Experiments

### 3.1 Evaluation Datasets and Metrics

We evaluate our MARS on several challenging benchmarks that require sophisticated multi-step reasoning and external knowledge: (1) **Humanity's Last Exam (HLE)** (Phan et al., 2025) is an extremely challenging dataset containing advanced problems. We utilize its text-only subset with 2,154 questions. (2) **Single-Hop Question Answering**, including NQ (Kwiatkowski et al., 2019), TriviaQA (Joshi et al., 2017), and PopQA (Mallen et al., 2023). (3) **Multi-Hop Question Answering**, including HotpotQA (Yang et al., 2018), 2WikiMultiHopQA (Ho et al., 2020), Musique (Trivedi et al., 2022), and Bamboogle (Press et al., 2023). For evaluation, we follow different protocols for different benchmarks. For HLE, we adopt its official evaluation prompt (Phan et al., 2025) with GPT-4o (Hurst et al., 2024) as the judge. For other QA datasets, following Chen et al. (2025), we employ Qwen2.5-72B-Instruct (Yang et al., 2024) as the evaluation model.

### 3.2 Baselines and Implementation Details

To evaluate the effectiveness of MARS, we compare our method with the following baselines. (1) **Direct Reasoning**: Models that directly answer questions without external knowledge, including open-source models (Qwen2.5 series (Yang et al., 2024) and QwQ-32B (Team, 2024)) and powerful proprietary models (DeepSeek-R1-671B (Guo et al., 2025), GPT-4o (Hurst et al., 2024), o1 (OpenAI, 2024), Claude 3.7 Sonnet (Anthropic, 2025), Gemini 2.5 Pro (Google, 2025a), o3(high) (OpenAI, 2025b)), and o4-mini(high) (OpenAI, 2025c). (2) **Advanced RAG Reasoning Methods**: We consider the standard RAG that retrieve top-10 documents based on the question, and several iterative RAG methods, including Self-RAG (Asai et al., 2024), InstructRAG (Wei et al., 2025), Auto-RAG (Yu et al., 2024b) and C-3PO (Chen et al., 2025). (3) **R1-like Reasoning with Search**: Methods that integrate external knowledge into R1-like reasoning, including Search-o1 (Li et al., 2025a), Search-R1 (Jin et al., 2025), WebThinker (Li et al., 2025b), and OpenAI Deep Research (OpenAI, 2025a).

We initialize our policy model with Qwen2.5-7B-Instruct (Yang et al., 2024) and Qwen3-8B (Yang et al., 2025) and design dedicated prompts for System 1 and System 2 (Appendix E) to facilitate multi-agent reinforcement learning. To ensure a fair comparison, we highlight that MARS adopts RL from zero without any SFT distillation. All methods are compared in similar public sources (Appendix C), ensuring data source parity. Additional implementation details are provided in Appendix D.

Table 1: Main Results on HLE (evaluated with official evaluation prompt (Phan et al., 2025) by GPT-4o). Results for proprietary models are from the official leaderboard for reference. For open-source models, the best results are in **bold** and the second results are underlined.

| Method | Humanity's Last Exam | | | | | | | | |
|---|---|---|---|---|---|---|---|---|---|
| | **Bio/Med** | **Chem.** | **CS/AI** | **Engineering** | **Humanities** | **Math** | **Physics** | **Other** | **Avg.** |
| *Proprietary Models (For Reference)* | | | | | | | | | |
| OpenAI Deep Research | - | - | - | - | - | - | - | - | 26.60 |
| o3 (high) | - | - | - | - | - | - | - | - | 20.57 |
| o4-mini (high) | - | - | - | - | - | - | - | - | 18.90 |
| Gemini 2.5 Pro | - | - | - | - | - | - | - | - | 18.38 |
| Deepseek R1 | - | - | - | - | - | - | - | - | 8.54 |
| Claude 3.7 Sonnet | - | - | - | - | - | - | - | - | 7.89 |
| o1 | - | - | - | - | - | - | - | - | 7.75 |
| GPT-4.1 | - | - | - | - | - | - | - | - | 4.91 |
| GPT-4o | - | - | - | - | - | - | - | - | 2.32 |
| *Open-Source Models* | | | | | | | | | |
| QwQ-32B | 9.05 | 6.00 | 4.86 | 1.61 | 6.21 | 4.92 | 4.95 | 3.42 | 5.28 |
| Qwen2.5-72B | 11.31 | 6.00 | 1.76 | 1.61 | 7.25 | 3.07 | 3.96 | 2.28 | 4.27 |
| Qwen2.5-7B | 5.42 | 3.00 | 1.76 | 3.22 | 4.66 | 3.58 | 1.98 | 4.00 | 3.52 |
| Qwen3-8B | 6.78 | 4.00 | 3.53 | 3.22 | 5.69 | 4.21 | 3.46 | 2.85 | 4.31 |
| *R1-like Reasoning with Search* | | | | | | | | | |
| WebThinker(QwQ-32B) | **14.47** | **8.00** | 4.42 | 6.45 | 10.88 | 4.51 | 1.98 | **14.28** | 6.87 |
| C-3PO (Qwen2.5-72B) | 9.95 | 7.00 | 4.86 | **9.67** | 4.66 | 5.43 | 3.46 | 5.71 | 5.79 |
| Search-o1(Qwen2.5-7B) | 9.95 | 2.00 | 2.67 | 1.61 | 4.14 | 4.41 | 3.98 | 7.42 | 4.79 |
| Search-R1(Qwen2.5-7B) | 6.33 | 6.00 | 4.42 | 6.45 | 3.62 | 3.59 | 1.48 | 4.00 | 3.99 |
| MARS (Qwen2.5-7B) | 12.66 | 3.00 | 5.75 | 4.83 | **11.92** | 6.46 | 6.43 | 7.42 | 7.38 |
| MARS (Qwen3-8B) | 13.12 | 6.00 | **8.84** | 6.45 | 8.29 | **7.17** | **7.92** | 8.57 | **8.17** |

## 3.3 Main Results on Humanity's Last Exam

Table 1 compares the performance of our MARS with various baselines on the Humanity's Last Exam (HLE) benchmark. This benchmark is particularly challenging as it contains advanced problems across multiple disciplines that require sophisticated reasoning and up-to-date knowledge. We have observed several key findings:

First, **MARS achieves an average accuracy of 7.38% across all categories, representing a substantial improvement of 3.86% over the base model (Qwen2.5-7B-Instruct)**. Notably, MARS outperforms all other open-source models and reasoning methods, including those based on much larger models such as WebThinker(QwQ-32B) and C-3PO(Qwen2.5-72B-Instruct). This demonstrates that our dual-system paradigm effectively enhances reasoning capabilities with significantly fewer parameters, achieving superior performance. Second, when examining subject-specific performance, MARS demonstrates particularly strong results in knowledge-intensive domains. This advantage stems from **our dual-system design, where System 1 efficiently processes and distills large volumes of retrieved information from web pages and research papers without consuming System 2's reasoning tokens.** This architecture allows System 2 to focus exclusively on complex reasoning while having access to comprehensive, filtered knowledge, which is especially beneficial for technical domains requiring both extensive background information and multi-step reasoning. Finally, while proprietary models like OpenAI Deep Research still maintain a lead in overall performance, MARS significantly narrows the gap between open-source and commercial solutions. **The performance gaps between our MARS (7.38%) and proprietary models like Claude 3.7 Sonnet (7.89%) or o1 (7.75%) is notably small, especially considering that MARS utilizes only a 7B parameter model.** These results demonstrate that our dual-system paradigm creates an effective synergy: System 1 efficiently filters and distills large volumes of external information, while System 2 maintains focused, deliberate reasoning without token consumption trade-offs.

Table 2: Main Results on Knowledge-intensive Tasks.

| Method | Single-Hop QA | | | Multi-Hop QA | | | | Avg. |
|---|---|---|---|---|---|---|---|---|
| | NQ | TriviaQA | PopQA | HotpotQA | 2Wiki | Musique | Bamboogle | |
| Direct Answer | 29.6 | 50.0 | 30.8 | 30.6 | 28.4 | 13.4 | 30.4 | 30.45 |
| Standard RAG | 45.6 | 72.0 | 46.2 | 43.4 | 29.4 | 25.6 | 41.6 | 43.40 |
| Self-RAG | 49.4 | 66.2 | 38.6 | - | - | - | - | 51.40 |
| InstructRAG | 47.8 | 66.6 | 39.6 | - | 35.8 | - | - | 47.45 |
| Auto-RAG | 52.4 | 62.2 | 36.8 | 44.4 | 46.8 | - | - | 48.52 |
| Search-o1 | 42.4 | 64.6 | 38.2 | 46.8 | 52.8 | 26.2 | 56.0 | 46.71 |
| Search-R1 | 51.6 | 72.6 | 56.8 | 46.6 | 50.4 | 28.4 | 57.4 | 51.97 |
| C-3PO | 47.4 | 75.0 | 56.8 | 48.2 | 52.6 | 33.2 | 60.8 | 53.42 |
| MARS (Qwen2.5-7B) | 60.6 | 76.4 | 64.4 | 60.4 | 66.4 | 39.6 | 68.8 | 62.37 |
| MARS (Qwen3-8B) | **63.2** | **78.6** | **66.8** | **63.6** | **69.2** | **42.4** | **71.2** | **65.00** |

## 3.4 MAIN RESULTS ON KNOWLEDGE-INTENSIVE REASONING

We present the results on 7 knowledge-intensive question answering tasks in Table 2. The evaluation covers both single-hop (NQ, TriviaQA, PopQA) and multi-hop (HotpotQA, 2Wiki, Musique, Bamboogle) datasets, revealing several key findings:

First, our MARS consistently outperforms all baseline methods across all benchmarks, achieving an impressive improvement of 8.95% over the previous state-of-the-art method C-3PO. This consistent performance gain across diverse datasets demonstrates the robustness and generalizability of our dual-system paradigm for knowledge-intensive reasoning tasks. Second, the performance gains are particularly pronounced on multi-hop reasoning tasks, where MARS achieves an average improvement of 12.2% over C-3PO across the four multi-hop benchmarks. This significant enhancement demonstrates that our approach excels at complex reasoning chains requiring multiple steps of information retrieval and integration. The multi-agent framework **allows System 2 to decompose complex questions into manageable sub-queries, while System 1 efficiently processes retrieved information, creating an effective synergy for multi-hop reasoning**. Third, even on relatively simpler single-hop QA tasks, MARS achieves an average gain of 7.5% over the previous best methods. This indicates that our dual-system approach enhances performance across the entire spectrum of reasoning complexity, from straightforward factual queries to intricate multi-step problems. By optimizing the collaboration between System 1's efficient information processing and System 2's deliberate reasoning capabilities, our MARS effectively leverages external knowledge sources while maintaining computational efficiency. These results further confirm that our approach mitigates the limitations of previous methods by **creating a more balanced and effective division of labor between intuitive and deliberate reasoning processes**.

## 3.5 ANALYSIS OF MULTI-AGENT RL PROCESS

To gain deeper insights into our multi-agent, multi-turn, multi-tool RL process, we conduct a comprehensive analysis of MARS's training dynamics. Figure 3 illustrates various aspects of the training progression, revealing several key patterns:

**First**, examining the core performance metrics, we observe a consistent and stable improvement in HLE score (Fig. 3a) from approximately 2% to over 10% on the HLE subset, demonstrating the effectiveness of our approach. This performance gain correlates with the training reward curve (Fig. 3b), which stabilizes around 0.4 after an initial increase. Interestingly, the number of tools used per question (Fig. 3c) also shows an upward trend, increasing from approximately 1 to over 2 by the end of training, indicating that MARS learns to leverage multiple tools for addressing complex questions. **Second**, the evolution of tool selection preferences (Fig. 3d-3f) reveals that while Google Search emerges as the predominantly chosen tool (reaching nearly 98% usage), our model maintains the capability to utilize Python and Google Scholar when appropriate. This balanced tool selection strategy allows System 2 to autonomously choose the most suitable tools for different question types,

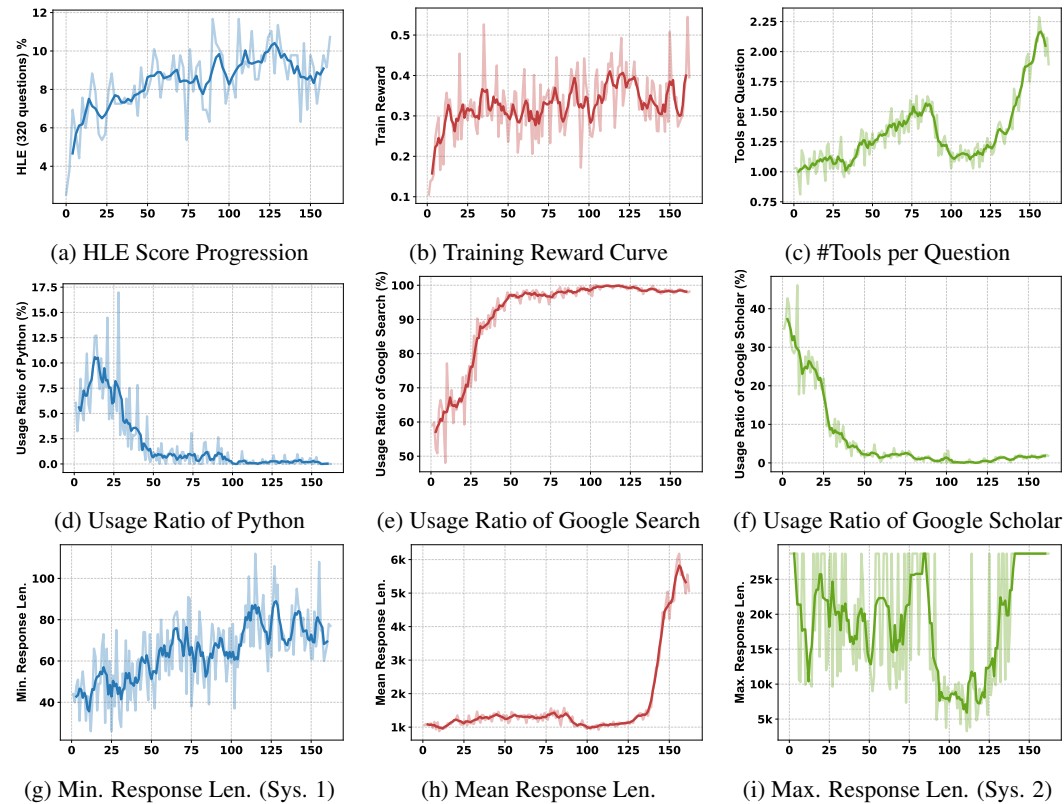

Figure 3: Comprehensive analysis of our RL training process. The $x$-axis represents training steps for all subfigures. (a-c) **Core performance metrics** on HLE score (randomly select 320 questions), training reward, and tool usage frequency per question. (d-f) Evolution of **tool selection preferences** across three available tools. While Google Search emerges as the predominantly chosen tool due to our training data distribution, we maintain all tools to preserve System 2's autonomous tool selection capability for diverse scenarios. (g-i) **Response length distributions** showing minimum (predominantly System 1), mean, and maximum (predominantly System 2) response lengths. Training was terminated after step 150 due to consistently exceeding our preset length constraints.

Table 3: Ablation Study on Tools for HLE.

| Tools | | | Humanity's Last Exam | | | | | | | | |
|:---:|:---:|:---:|:---:|:---:|:---:|:---:|:---:|:---:|:---:|:---:|:---:|
| 🐍 Python | G Search | 🔵 Scholar | Bio/Med | Chem. | CS/AI | Engineering | Humanities | Math | Physics | Other | Avg. |
| ✓ | ✓ | ✓ | 12.66 | 3.00 | 5.75 | 4.83 | **11.92** | **6.46** | **6.43** | 7.42 | **7.38** |
| ✗ | ✓ | ✓ | **13.12** | 5.00 | 5.31 | 4.83 | 8.29 | 4.92 | 5.45 | 5.14 | 6.21 |
| ✓ | ✗ | ✓ | 11.31 | 6.00 | **7.07** | 3.22 | 6.73 | 4.92 | 3.47 | 6.85 | 5.99 |
| ✓ | ✓ | ✗ | 11.76 | **7.00** | 4.42 | 4.83 | 8.81 | 6.05 | 4.45 | **8.00** | 6.72 |
| ✓ | ✗ | ✗ | 4.54 | 4.00 | 3.98 | **6.45** | 4.14 | 5.02 | 3.96 | 4.57 | 4.64 |
| ✗ | ✓ | ✗ | 12.66 | 6.00 | 5.31 | 3.22 | 7.77 | 4.41 | 5.94 | **8.00** | 6.12 |
| ✗ | ✗ | ✓ | 12.21 | **7.00** | 4.42 | 4.83 | 5.69 | 4.51 | 5.44 | 4.57 | 5.61 |

enhancing the model's versatility across diverse reasoning scenarios. **Third**, the response length dynamics (Fig. 3g-3i) provide valuable insights into the operation of our dual-system architecture. The minimum response length (mostly System 1) and maximum response length (mostly System 2) both show increasing trends during training, suggesting that both systems learn to generate more comprehensive and informative outputs. The significant increase in mean response length indicates that MARS progressively develops more sophisticated reasoning capabilities as training advances.

### 3.6 ABLATION STUDY

To further investigate the impact of different external tools on reasoning performance, we conduct a comprehensive ablation study on the HLE benchmark, as shown in the Table 3.

We first observe that the model with all three tools achieves the best overall performance (7.38%), confirming that the combination of computational capabilities and diverse knowledge sources provides the most robust foundation for complex reasoning. Interestingly, the performance drop varies significantly depending on which tool is removed, revealing their differential importance across subject domains: removing Python affects Math (-1.54%) and Physics (-0.98%) most severely, while actually improving performance in Bio/Med (+0.46%) and Chemistry (+2.00%). Second, **Google Search proves to be the most versatile single tool**, with its removal causing the largest overall performance drop (1.38%), particularly in Physics (-2.95%). Then, **Google Scholar** contributes most significantly to CS/AI and Other categories, **reflecting its value for domains with rapidly evolving research**. These findings demonstrate that **different tools contribute uniquely across subject domains**. The complementary nature of these tools enables MARS to adapt its reasoning strategy based on the specific requirements of each question, highlighting the importance of our multi-tool approach in addressing the diverse challenges presented by complex reasoning tasks.

### 3.7 INFORMATION VOLUME ANALYSIS

In our MARS, each tool invocation supports multiple queries, with each query retrieving up to 10 complete web pages or 5 full academic papers, resulting in substantial information volume that System 1 effectively processes and distills for System 2's reasoning. As

Table 4: Information Volume per Question

| Dataset | G Web Pages | Papers | Python |
|---------|-------------|--------|--------|
| PopQA | 18.81 | 0.04 | 0.0 |
| hotpotqa | 17.38 | 0.12 | 0.002 |
| HLE | **22.31** | **0.17** | **0.05** |

shown in Table 4, MARS retrieves and processes a significant amount of information per question across all benchmarks. Particularly for HLE, the model accesses an average of 22.31 web pages and 0.17 academic papers per question, reflecting the complexity and knowledge-intensive nature of this benchmark. This analysis demonstrates that MARS effectively leverages a large volume of external information, highlighting the efficiency of our dual-system approach in processing and utilizing retrieved knowledge.

## 4 RELATED WORK

**Retrieval-Augmented Generation (RAG)** has emerged as a crucial approach to overcome knowledge limitations of large language models by integrating external information sources (Lewis et al., 2020; Nakano et al., 2021; Schick et al., 2023; Yu et al., 2024c; Asai et al., 2024; Jiang et al., 2024; Wei et al., 2025; Yu et al., 2024b; Chen et al., 2025; Li et al., 2025b). However, existing RAG systems typically struggle with either information overload when processing multiple lengthy documents (such as entire web pages or research papers) or loss of critical details when condensing information (Fan et al., 2024; Gao et al., 2023; Tan et al., 2024). Our approach addresses these limitations through a dual-system framework where System 2 handles complex reasoning while System 1 efficiently processes retrieved information, enabling MARS to manage larger volumes of external information while maintaining reasoning quality across diverse knowledge-intensive tasks. Due to space constraints, additional related work is provided in Appendix B.

## 5 CONCLUSION

In this paper, we presented MARS, a multi-agent system for Deep Research that seamlessly integrates System 1's fast, intuitive thinking with System 2's deliberate reasoning. By creating a specialized division of labor between the two cognitive systems, MARS establishes an efficient collaborative framework where each component operates within its optimal domain. Furthermore, we proposed a multi-agent RL framework to simultaneously optimize both systems with bin-packing optimization and sample balancing strategies to enhance the collaborative efficiency. Extensive experiments demonstrate MARS's superior performance and the effectiveness of our dual-system paradigm for complex reasoning tasks that require processing large volumes of external information.

ETHICS STATEMENT

This work adheres to the ICLR Code of Ethics. Our research does not involve human subjects, does not raise privacy or security concerns, and does not have potential negative societal impacts. We have no conflicts of interest to declare.

REPRODUCIBILITY STATEMENT

To ensure the reproducibility of our work, we have made significant efforts to provide comprehensive details and resources. The experimental setup, including detailed descriptions of all datasets used, is presented in Section 3 and Appendix D. Implementation details, including hyperparameters, training procedures, and data construction methods, are thoroughly documented in Appendix D. We provide complete source code and all processed datasets as supplementary materials, which include step-by-step instructions for reproducing all experimental results reported in this paper. The supplementary materials contain: (1) all source code with documentation, (2) scripts for data preprocessing and construction, (3) detailed README files explaining how to run the experiments. All experiments were conducted using publicly available libraries, and we specify the exact versions used to ensure consistent reproduction of our results.

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

## A   LLM Usage Statement

We acknowledge the use of Large Language Models (LLMs) to assist with improving the grammar, clarity, and readability of this manuscript. The LLMs were employed solely for language polishing purposes, including refining sentence structures, correcting grammatical errors, and enhancing the overall flow of the text. All research ideas, experimental design, technical content, and scientific contributions are entirely the work of the human authors. We have carefully reviewed and verified all LLM-suggested edits, and take full responsibility for the accuracy and integrity of the final manuscript.

## B   Additional Related Work

**Language Reasoning Models (LRMs).**   Recent advances in language models have witnessed the emergence of LRMs, which specifically excel at deliberate, System 2-type thinking (Jaech et al., 2024; Yu et al., 2024a; Guo et al., 2025; Team, 2024; Anthropic, 2025; Li et al., 2025c; Ziabari et al., 2025). Despite these advances, current LRMs rely on static, parameterized knowledge acquired during pre-training, without access to external world information (Chen et al., 2025; Li et al., 2025a; Jin et al., 2025; Li et al., 2025b). Furthermore, when faced with simpler tasks, such as analyzing web pages or reading papers, LRMs may tend to overanalyze, consuming excessive reasoning tokens that limit their capacity to process large volumes of information. Our work addresses these limitations by integrating System 1's fast, intuitive thinking with System 2's deliberate reasoning in a multi-agent framework that leverages external knowledge sources.

**Multi-agent Systems.**   Multi-agent systems have gained significant attention in the LLM community as a means to tackle complex tasks through collaborative problem-solving (Park et al., 2023; Wu et al., 2023; Li et al., 2023; Hong et al., 2023; Qian et al., 2023; Han et al., 2024; Li et al., 2024; Chen et al., 2025). Recent work has explored various agent architectures, from simple role-playing approaches (Li et al., 2023; Du et al., 2023) to more sophisticated frameworks with specialized agents handling different aspects of complex tasks (Hong et al., 2023; Wang et al., 2023). However, most existing multi-agent systems employ agents with similar cognitive architectures, typically all operating in a intuitive, System1-type thinking or deliberate, System 2-like reasoning mode. This homogeneity limits their ability to efficiently process large volumes of information while maintaining complex reasoning capabilities. Our approach diverges by implementing a heterogeneous multi-agent system where System 1 and System 2 agents possess complementary cognitive abilities, optimized through reinforcement learning (Yuan et al., 2023; Zhu et al., 2024; Shao et al., 2024) to collaboratively tackle complex research tasks requiring both efficient information processing and deliberate reasoning.

## C   Training Data Filtering Pipeline

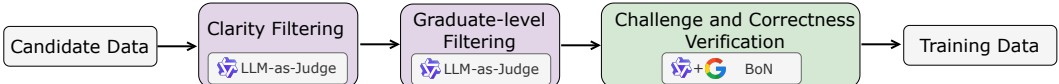

Figure 4: Our data curation pipeline.

Given the absence of specialized training datasets for enhancing model capabilities required for challenging benchmarks such as HLE (Phan et al., 2025) (which demands web browsing, reasoning, and computation skills), we propose a data curation pipeline that selectively identifies suitable training examples from open-source datasets. As illustrated in Figure 4, our pipeline evaluates candidate examples based on three key dimensions: clarity of presentation, problem complexity, and solution correctness.

All candidate data are sourced from publicly available datasets, including MMIQC Liu et al. (2025), WebInstructSub Yue et al. (2024), GeneralThought[2], and SuperGPQA Du et al. (2025). To ensure quality for our dual-system training, we implement a rigorous multi-stage filtering process:

---

[2]https://huggingface.co/datasets/GeneralReasoning/GeneralThought-430K

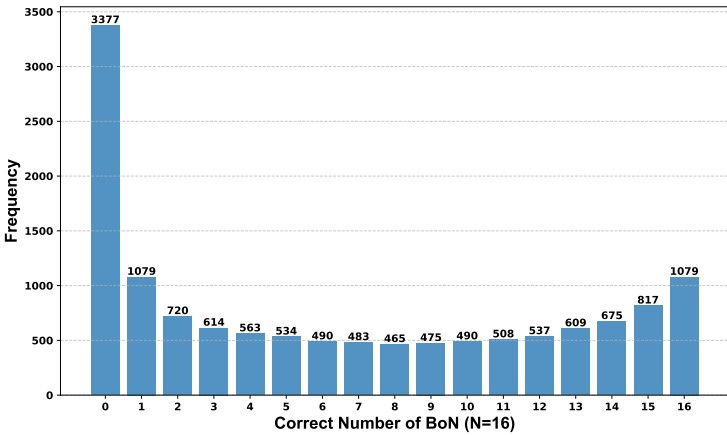

Figure 5: Distribution of Correct Number in Best-of-N ($N = 16$). Questions answered correctly 1-12 times were retained for training, while those answered 0 times (potentially ambiguous or lacking definitive solutions) or ¿12 times (too trivial) were excluded.

1. **Initial filtering by academic level:** Starting from an initial pool of 5 million examples, we categorize each prompt by academic discipline and retain only those that may meet undergraduate or graduate-level difficulty standards, reducing the dataset to 237K examples.
2. **Deduplication:** We remove near-identical prompts that differ only in phrasing or superficial elements, yielding 155K unique prompts.
3. **Clarity assessment:** We employ an LLM (Qwen2.5-72B-Instruct (Yang et al., 2024)) to systematically evaluate the clarity of each prompt, filtering out those deemed ambiguous or poorly formulated, resulting in 99K high-quality prompts.
4. **Graduate-level Filtering:** We further use the same LLM to assess the challenge level of each prompt, retaining only those that meet graduate-level standards in their respective disciplines, resulting in 81K prompts of appropriate difficulty.
5. **Challenge and Correctness Verification:** To ensure both appropriate difficulty and answer verifiability, we perform a best-of-16 (BoN) sampling procedure using Qwen2.5-72B-Instruct with Google Search access. As shown in Figure 5, we plot the distribution of how many times (out of 16 attempts) the model correctly answers each question. We retain only questions with moderate difficulty (correctly answered 1-12 times out of 16 attempts), which serves two critical purposes: (1) eliminating trivial questions (answered correctly ¿12 times) and excessively difficult ones (never answered correctly), and (2) identifying questions with consistent, verifiable answers, as those never answered correctly may lack definitive solutions or contain ambiguous information.
6. The final dataset comprises 40K carefully curated prompts spanning diverse academic disciplines and difficulty levels, specifically designed to support high-quality dual-system training for complex reasoning tasks.

To facilitate research advancement and reproducibility, we have open-sourced our curated 40K dataset. The detailed prompts used throughout our data curation pipeline are provided in Appendix E.2.

## D    More Implementation Details

In this section, we provide a comprehensive implementation details of our proposed method. For additional insights and more intricate details, we refer the reader to our supplementary materials.

### D.1    Overall Algorithm

Our dual-system approach leverages the complementary strengths of two distinct cognitive systems within the same large language model. Algorithm 1 presents the complete multi-agent rollout process that orchestrates the interaction between System 1 (fast, intuitive processing) and System 2 (deliberate, analytical reasoning).

---

**Algorithm 1** Multi-agent Rollout Process

---

**Require:** Question $q$, System 1 $\pi_{\text{sys}_1}$ and System 2 $\pi_{\text{sys}_2}$ in the same LLM, candidate tools, Maximum interaction turns $N_{\max}$.

1:  $i \leftarrow 0$                                                  ▷ *Initialize interaction turns*

2:  $c_0 \leftarrow \{q\}$                            ▷ *Initialize reasoning context for System 2*

3:  $c_{\text{sys}_1} \leftarrow \emptyset$                       ▷ *Initialize collection of System 1 input-output pairs*

4:  **while** $i < N_{\max}$ **do**

5:     $s_i, (t_i, p_i) \leftarrow \pi_{\text{sys}_2}(c_i)$ ▷ *System 2 generates reasoning step $s_i$, (optional tool request $t_i$ and purpose $p_i$)*

6:     **if** $t_i$ and $p_i$ are not Empty **then**                   ▷ *Tool request present*

7:         $\{o_{t_i}^{(1)}, o_{t_i}^{(2)}, \ldots, o_{t_i}^{(n_{t_i})}\} \leftarrow$ Tool Call$(t_i)$         ▷ *Execute tool call*

8:         $\{b_1, \ldots, b_m\} \leftarrow$ Bin-Packing$(o_{t_i}^{(1)}, \ldots, o_{t_i}^{(n_{t_i})})$         ▷ *m chunks*

9:         $\{d_1, \ldots, d_m\} \leftarrow \{\pi_{\text{sys}_1}(b_1), \ldots, \pi_{\text{sys}_1}(b_m)\}$   ▷ *Parallel distillation (System 1)*

10:        $c_{\text{sys}_1} \leftarrow c_{\text{sys}_1} \cup \{(b_j, d_j)\}_{j=1}^m$         ▷ *Collect all input-output pairs*

11:        $\tilde{o}_{t_i} \leftarrow \cup_{j=1}^m d_j$                   ▷ *Combine all distilled information*

12:        $c_{i+1} \leftarrow c_i \cup \{s_i, t_i, p_i, \tilde{o}_{t_i}\}$     ▷ *Update System 2's context with tool results*

13:     **else**

14:        $c_{i+1} \leftarrow c_i \cup \{s_i\}$                  ▷ *Update context with reasoning only*

15:        **if** `<answer> </answer>` in $s_i$ **then**      ▷ *Check if answer is provided*

16:            **break**

17:     $i \leftarrow i + 1$

18: **return** Rollout trajectory $c_i$ and System 1 input-output pairs $c_{\text{sys}_1}$

---

Inspired by human cognition, our MARS assigns distinct roles to each system: System 2 manages strategic reasoning and decision-making, while System 1 efficiently processes and distills large volumes of information. The algorithm enables System 2 to independently continue reasoning when no tool is required, and terminates either when an answer is reached or after the maximum number of turns is exhausted. The parallel processing capability of System 1 is particularly valuable when handling extensive tool-retrieved information. By distributing information processing across multiple parallel instances of System 1, our approach efficiently manages complex information needs without exceeding context window limitations.

### D.2 BIN-PACKING DETAILS

As mentioned in Section 2.2.1, we employ a First Fit Decreasing (FFD) algorithm to efficiently organize variable-length tool outputs into optimally-sized chunks. Our bin-packing implementation follows these key steps:

1. **Token counting**: For each tool output $o_{t_i}^{(j)}$, we count the number of tokens using the model's tokenizer.

2. **Large output handling**: If any single output exceeds the maximum context length of System 1, it is truncated and placed in a dedicated bin.

3. **Sorting**: Remaining outputs are sorted in decreasing order of their token lengths.

4. **Bin assignment**: Each output is assigned to the first bin that can accommodate it without exceeding the context length limit. If no existing bin has sufficient space, a new bin is created.

### D.3 REWARD DESIGN

We introduce a straightforward yet effective reward design for our multi-agent RL training. For each rollout trajectory, we employ LLMs (Qwen2.5-72B-Instruct) as evaluators to assess the correctness of the predicted final answer following HLE's official prompt (Phan et al., 2025). The reward function is defined as:

$$r(c_N, \text{ground truth}) = \begin{cases} 1, & \text{if Eval}_{\text{LLM}} = \text{Correct} \\ 0, & \text{otherwise} \end{cases} \tag{8}$$

where $c_N$ is the final reasoning trajectory, $N$ is the number of interaction turns. We will extract the answer part between <answer> and </answer> tags for evaluation.

All System 1 and System 2 samples within the same trajectory share this trajectory-level reward. This design reflects the inherently collaborative nature of our dual-system framework, where the final answer quality depends on both System 2's reasoning and System 1's information processing. By sharing rewards, we encourage both systems to optimize toward the same goal—producing correct final answers—rather than pursuing potentially conflicting individual objectives. This binary reward signal creates a clear learning objective for both systems: System 2 learns to generate better reasoning steps and more effective tool-use plans, while System 1 learns to distill information more accurately and concisely to support System 2's reasoning. The simplicity of this reward function helps avoid the common pitfalls of overly complex reward engineering while maintaining focus on the ultimate goal of correct problem-solving.

### D.4 RL Loss

We write the policy loss of one single rollout trajectory as follows.

$$\mathcal{L}_{\text{policy}}(x, y, A_{\text{sys}_i}) = \frac{1}{|y|} \sum_{j=1}^{|y|} \min \left[ \frac{\pi_{\text{sys}_i}(y_j|x, y_{<j})}{\pi_{\text{sys}_i}^{\text{old}}(y_j|x, y_{<j})} A_{\text{sys}_i}, \text{clip} \left( \frac{\pi_{\text{sys}_i}(y_j|x, y_{<j})}{\pi_{\text{sys}_i}^{\text{old}}(y_j|x, y_{<j})}, 1 - \epsilon, 1 + \epsilon \right) A_{\text{sys}_i} \right]$$

(9)

where $y_j$ is the $j$-th token of LLM output $y$. Similarly, we write the KL loss as follows.

$$\mathcal{L}_{\text{KL}}(x, y) = -\frac{1}{|y|} \sum_{j=1}^{|y|} \mathbb{D}_{\text{KL}} \left( \pi_{\text{sys}_i}(y_j|x, y_{<j}) \| \pi_{\text{sys}_i}^{\text{ref}}(y_j|x, y_{<j}) \right)$$

(10)

In practical implementation, the expectation in Eq. (7) is achieved via averaging over a group of $G$ rollouts as well as a batch training examples.

### D.5 Implementation Details

Table 5: Key hyperparameters in the RL phase.

| Hyperparameter | Value |
|---|---|
| Learning Rate of Policy model | 1e-6 |
| Base model | Qwen2.5-7B-Instruct |
| Batch size | 32 |
| $G$ | 16 |
| temperature | 1.0 |
| KL loss coefficient $\lambda$ | 0. |
| entropy coefficient | 0. |
| Maximum Prompt Length of System 1 | 23,552 |
| Maximum Response Length of System 1 | 8192 |
| Maximum Prompt Length of System 2 | 3072 |
| Maximum Response Length of System 2 | 28,672 |
| Maximum interaction turns | 10 |

This section provides comprehensive implementation details of our MARS. We initialize our policy model with Qwen2.5-7B-Instruct (Yang et al., 2024), which serves as the foundation for both System 1 and System 2. Furthermore, we provide external tools such as Google Search, Google Scholar via the SerpAPI[3], and Python code interpreter for supporting autonomous tool selection during the reasoning process. We employ the GRPO algorithm (Shao et al., 2024) with the following hyperparameters: learning rate of 1e-6, batch size of 32, sampled responses per prompt (group size $G$) of 16, temperature of 1.0, top-p of 0.95, and both KL loss coefficient and entropy coefficient set to 0. Additionally, we set different maximum lengths for System 1 and System 2: prompt lengths

---

[3]https://serpapi.com/

of 23,552 and 3,072 tokens, and response lengths of 8,192 and 28,672 tokens, respectively. This configuration enables System 1 to better incorporate external knowledge while allowing System 2 to focus on sophisticated multi-step reasoning. For all baselines, we maintain their original settings in their respective papers to ensure optimal performance. Table 5 summarizes the key hyperparameters used during the reinforcement learning phase.

The asymmetric prompt and response length configurations between System 1 and System 2 are designed to leverage their complementary roles. System 2, with its shorter prompt length (3,072 tokens) but longer response capability (28,672 tokens), is optimized for detailed reasoning and solution generation. Conversely, System 1, with its extended prompt length (23,552 tokens) but more concise response limit (8,192 tokens), excels at processing and summarizing large volumes of information. This configuration aligns with cognitive science theories where System 2 handles deliberate reasoning while System 1 processes information rapidly.

### D.6 BASELINE COMPARISONS

For all baseline models, we made every effort to ensure fair comparison by using their original implementations and settings. However, readers may notice that our reported results for WebThinker (QwQ-32B) in Table 1 differ from those in the original paper (Li et al., 2025b). This discrepancy stems from two differences:

1. **Evaluation scale**: Our evaluation encompasses the complete HLE text-based question set (2,154 questions), providing a more comprehensive assessment than the 500-question sample used in the original WebThinker paper.
2. **Evaluation criteria**: We strictly adhere to HLE's official evaluation prompt and criteria for all models, including WebThinker. In contrast, the original WebThinker paper employed its own evaluation prompt that resulted in more lenient scoring.

### D.7 DATASET DETAILS

Our training data consists of two complementary components: (1) a curated collection of complex reasoning examples filtered through our specialized pipeline, as shown in Appendix C, and (2) several established open-source single-hop, multi-hop, and Biology & Medicine datasets that enhance the model's knowledge retrieval and reasoning capabilities. Table 6 summarizes the key statistics of our complete training dataset.

Table 6: Data Statistics.

| Data Type Data Name | Our curated data | Single-Hop QA | | Multi-Hop QA | | | Biology&Medicine | | Total |
|---|---|---|---|---|---|---|---|---|---|
| | | TriviaQA | PopQA | HotpotQA | 2Wiki | Musique | PubMedQA | CUPCase | |
| Sampled Number | 2000 | 400 | 400 | 500 | 500 | 500 | 500 | 250 | 5050 |

As shown in Table 6, we randomly sampled a total of 5,050 training examples across eight distinct datasets. The composition is carefully balanced to ensure comprehensive coverage of different reasoning types and knowledge domains.

- **Our curated data (2,000 examples, 39.6%):** These examples were selected from our pipeline-filtered corpus described in Appendix C, which contains high-quality, graduate-level complex reasoning tasks vetted through our rigorous multi-stage filtering process.
- **Single-Hop QA (800 examples, 15.8%):** We incorporated 400 randomly sampled examples each from TriviaQA (Joshi et al., 2017) and PopQA (Mallen et al., 2023). These datasets focus on direct factual knowledge retrieval, with TriviaQA covering a broad range of trivia questions and PopQA specifically targeting popular entities and common knowledge.
- **Multi-Hop QA (1,500 examples, 29.7%):** To strengthen the model's multi-step reasoning capabilities, we included 500 examples each from HotpotQA (Yang et al., 2018), 2WikiMultihopQA (Ho et al., 2020), and MuSiQue (Trivedi et al., 2022). These datasets require reasoning across multiple documents or knowledge pieces to arrive at the correct answer.
- **Biology & Medicine (750 examples, 14.9%):** To enhance domain-specific knowledge, we sampled 500 examples from PubMedQA (Reese et al., 2024) and 250 examples from CUPCase (Perets et al., 2025), covering biomedical research questions and clinical case analysis respectively.

## D.8 EXPERIMENT ENVIRONMENTS

All experiments were conducted on Ubuntu 22.04 equipped with NVIDIA A100 GPUs. Our implementation relies on Python 3.10[4] and PyTorch 2.6.0[5], while extending VeRL[6] for our multi-agent reinforcement learning framework. For efficient execution, we implemented rollout procedures based on Qwen-Agent[7] and use vLLM[8] and SGLang[9] as our inference engines. We use Code Sandbox[10] for Python Interpreter.

# E INSTRUCTION TEMPLATES

## E.1 INSTRUCTION OF SYSTEM 1 & 2 IN OUR MARS

---

**Instruction for System 1**

**¡System Prompt¿**
You are an expert information extractor. Your sole task is to extract only the information that directly supports the tool call's purpose or answers the user's question.

## Task Guidelines
1. **Match Each Query**: For every query, extract information directly relevant to it and record its source (e.g., title, section name).
2. **Content Scanning**: Locate the **specific sections/data** directly related to the user's goal within the content.
3. **Key Extraction**: Identify and extract the **most relevant information** from the content, you never miss any important information
4. **Verbatim Key Content**: Preserve the original wording of key definitions, claims, formulas, data points.
5. **Preserve Detail**: Include relevant data, numbers, metrics, or formulas.
6. **Output Structure**: Organize the extracted content per query in a clear and nested way.

**¡User Prompt¿**
{tool outputs}

# User Question: {question}

---

**Instruction for System 2**

**¡System Prompt¿**
You are an expert researcher who combines rigorous analytical reasoning with thorough information seeking abilities. You excel at solving complex problems through logical thinking, careful analysis, and responsible tool use. You are known for your careful and thorough approach, never rushing to conclusions without complete analysis.

{tool description}

When performing a search:
1. **Persistent Actions for Answers**: You can engage in multiple search iterations, delving deeply into the topic to explore all possible aspects until a satisfactory answer is found.
2. **Repeated Verification**: Before presenting a Final Answer, you will **cross-check**

---

[4] https://www.python.org/
[5] https://pytorch.org/
[6] https://github.com/volcengine/verl
[7] https://github.com/QwenLM/Qwen-Agent
[8] https://github.com/vllm-project/vllm
[9] https://github.com/sgl-project/sglang
[10] https://github.com/bytedance/SandboxFusion

and **validate the information** you've gathered to confirm its accuracy and reliability.
3. **Attention to Detail**: You will carefully analyze each information source to ensure that all data is current, relevant, and from credible origins.

Your reasoning process should be enclosed within ¡think¿ ¡/think¿ tags. If you need external support, make tool calls inside ¡tool_call¿ ¡/tool_call¿ tags. After a tool call, always reassess the result critically and continue your analysis in a new ¡think¿ section. Tools are helpful but not always reliable — treat their output with scrutiny.

Finally, present your key reasoning and final answer inside ¡answer¿ ¡/answer¿ tags.
Do not nest tags. Each tag block must be independent.

¡User Prompt¿
{question}

## E.2 INSTRUCTION FOR OUR DATA CURTION PIPELINE

In our data curation pipeline, we employed specific instructions for each filtering stage to ensure consistent evaluation criteria. Below, we provide the detailed prompts used for Clarity Filtering and Graduate-level Filtering stages, which were directly provided to Qwen2.5-72B-Instruct for assessment.

For the Challenge and Correctness Verification stage (Best-of-16 sampling), we utilized a different approach. Rather than using a standalone evaluation prompt, we leveraged our dual-system framework itself, combining the System 1 and System 2 instructions (detailed in Appendix E) with Google Search integration. This allowed Qwen2.5-72B-Instruct to perform complete reasoning attempts on each question, providing a more authentic assessment of difficulty level and answer verifiability than a static evaluation would allow.

---

**Instruction of Clarity Filtering**

### Instruction ###
You are a domain expert proficient in various subjects such as Math, Physics, Biology, Humanities, Computer Science, Engineering, and Chemistry. You will be given a question and its corresponding answer. Please identify whether the given QA-pair exhibits "Clarity". Only output 1 for yes and 0 for no.

### Definition ###
"Clarity" is defined as the question being clearly and unambiguously stated, and the answer being unique.

### Examples ###
{Few-shot Examples}

### QA-pair ###
{Input QA-pair}

### Your Judgment ###

---

**Instruction of Graduate-level Filtering**

### Instruction ###
You are a domain expert proficient in various subjects such as Math, Physics, Biology, Humanities, Computer Science, Engineering, and Chemistry. You will be given a question and its corresponding answer. Please identify whether the given QA-pair contains "Expert-level Knowledge." Only output 1 for yes and 0 for no.

---

### Definition ###
"Expert-level Knowledge" is defined as core theories, cutting-edge research, and complex applications studied during advanced undergraduate and graduate levels, typically mastered only by experts or senior researchers in the field.

### Examples ###
{Few-shot Examples}

### QA Pair ###
{Input QA-pair}

### Your Judgment ###

### E.3 INSTRUCTION FOR REWARD MODEL

For our reward model, we strictly adhere to the official evaluation criteria from the Humanity's Last Exam (HLE) benchmark (Phan et al., 2025).

---

**Instruction of Reward Model (official Instruction from HLE)**

Judge whether the following [response] to [question] is correct or not based on the precise and unambiguous [correct_answer] below.

[question]: {question}

[response]: {response}

Your judgement must be in the format and criteria specified below:

extracted_final_answer: The final exact answer extracted from the [response]. Put the extracted answer as 'None' if there is no exact, final answer to extract from the response.

[correct_answer]: {correct_answer}

reasoning: Explain why the extracted_final_answer is correct or incorrect based on [correct_answer], focusing only on if there are meaningful differences between [correct_answer] and the extracted_final_answer. Do not comment on any background to the problem, do not attempt to solve the problem, do not argue for any answer different than [correct_answer], focus only on whether the answers match.

correct: Answer 'yes' if extracted_final_answer matches the [correct_answer] given above, or is within a small margin of error for numerical problems. Answer 'no' otherwise, i.e. if there if there is any inconsistency, ambiguity, non-equivalency, or if the extracted answer is incorrect.

confidence: The extracted confidence score between 0—%— and 100—%— from [response]. Put 100 if there is no confidence score available.

