# OpenReview forum: "MARS: Optimizing Dual-System Deep Research via Multi-Agent Reinforcement Learning"
_ICLR.cc/2026/Conference — Submitted to ICLR 2026_

### Official Review · Reviewer_U9pi · 2025-11-01

**Soundness:** 2
**Presentation:** 2
**Contribution:** 2
**Rating:** 2
**Confidence:** 3

**Summary:**

Inspired by the dual-process theory of human cognition, this paper proposes a collaborative two-agent framework in which one agent (System 2) conducts deliberate reasoning and invokes external tools when necessary, while the other agent (System 1) summarizes and distills the tool outputs into concise feedback for the reasoning agent. The authors extend Group Relative Policy Optimization (GRPO) into a multi-agent reinforcement learning (RL) setting, incorporating bin-packing optimization and balanced sampling to improve learning efficiency and stability.

**Strengths:**

The proposed framework enhances reasoning through multiple rounds of tool usage and effective cooperation between the two systems. The results demonstrate consistent improvements over several baselines and achieve an HLE score comparable to large proprietary models such as Claude 3.7 Sonnet.

**Weaknesses:**

1. Writing quality: The exposition can be improved for clarity and coherence. The intended connection to the dual-process theory (System 1 vs. System 2) is not immediately clear and is somewhat confusing in the abstract.
2. Novelty concerns: The idea of using tool calls (e.g., retrieval, computation) has been explored extensively in prior work (e.g., WebGPT). Likewise, the computational optimization via bin-packing is a known technique rather than a novel contribution.
3. Baseline limitations: Although the method is positioned as a multi-agent system, most comparisons are against single-agent RAG or R1-style baselines. Including comparisons with multi-agent frameworks (e.g., CAMEL, and MetaGPT) would strengthen the empirical evaluation.

**Questions:**

1. Are System 1 and System 2 implemented and trained using the same policy LLM but differentiated only by prompts or roles? If so, why is this setup referred to as “multi-agent”?
2. The reported HLE result for WebThinker appears inconsistent with the original paper [1]; could the authors clarify this discrepancy?
3. What causes the increase in the number of tools per question as training progresses? Why does the agent predominantly use Google Search after approximately 50 training steps? What would happen if we only use Google search as the tool?
5. The mean response length increases sharply near the end of training, while the HLE score drops. How do the authors interpret this inverse correlation?

## [Reference]
[1] Li, Xiaoxi, et al. "Webthinker: Empowering large reasoning models with deep research capability." arXiv preprint arXiv:2504.21776 (2025).

---

> ### Author Response · Authors · 2025-11-16
>
> We sincerely thank the reviewer for their time and detailed, constructive feedback on our paper. We appreciate the opportunity to address the weaknesses and questions raised. Below, we provide point-by-point responses that we hope will clarify our contributions and the design choices of our framework.
>
> > Response to W1
>
> We thank the reviewer for their valuable feedback regarding the clarity of our writing, specifically concerning the connection between our framework and the dual-process theory in the abstract. We agree that a precise mapping is crucial for understanding our contribution.
>
> Our framework, MARS, implements the dual-system theory through a concrete and specialized division of labor between two agents, which we detail in Section 2.1 (Lines 111-122) and define via distinct prompts in Appendix E.1 (Lines 1039 & 1066):
>
> * System 2 (Deliberate Reasoning): This agent embodies the "slow, deliberate" thinking process. It serves as the core Reasoning Agent that analyzes the problem, performs multi-step reasoning, formulates plans, and strategically decides when to invoke external tools (e.g., Google Search, Python).
>
> * System 1 (Fast, Intuitive Processing): This agent handles the "fast, perceptual" load. It functions as an Information Processing Agent. When System 2 calls a tool, System 1 receives the high-volume, often noisy, raw output (e.g., multiple full web pages or papers). Its sole task is to rapidly distill and summarize this information into concise, key insights, which are then passed back to System 2.
>
> This division of labor directly mimics the cognitive theory: System 2 focuses on complex reasoning without being overwhelmed, while System 1 efficiently processes vast external stimuli (the tool outputs) into a manageable format for System 2's context.
>
> We believe this clarification will resolve the confusion noted by the reviewe
>
> > Response to W2
>
>
> We thank the reviewer for raising this important point about novelty. We agree that the individual components—tool use (as in WebGPT) and bin-packing algorithms—are well-established techniques.
>
> Our novelty, however, lies not in inventing these components, but in our new architecture and optimization framework that integrates them to solve a key challenge in advanced reasoning: information overload.
>
> 1. Novelty in Tool Use (vs. WebGPT):
>
> The reviewer correctly identifies that prior work like WebGPT uses tool calls. However, these systems typically use a single agent that is responsible for both reasoning and processing the full, raw, retrieved information. This creates a well-known bottleneck: the agent's limited context window is quickly overwhelmed by large volumes of text (e.g., multiple full webpages).
>
> Our **key innovation is the explicit division of labor** (as clarified in our response to W1) between two distinct agents, optimized via a Multi-Agent RL (MARL) framework:
>
> * System 2 (Reasoning Agent): Performs deliberate, R1-style reasoning, but it does not process the raw tool output. It only generates the purpose of the tool call.
>
> * System 1 (Information Processing Agent): This is a new, specialized agent whose sole task is to receive the massive, high-volume raw output (as shown in Table 4, an avg. of 22.3 webpages per question on HLE) and distill/summarize it for System 2 based on its "purpose".
>
> This dual-agent architecture directly addresses the information overload bottleneck that single-agent systems face. Our primary contribution is **the MARL framework (Section 2.2)** we propose to jointly optimize this complex S1-S2 collaboration, which is a significant step beyond standard RAG or single-agent tuning.
>
> 2. Role of Bin-Packing (An Enabling Optimization):
>
> Regarding bin-packing, we fully agree with the reviewer that the First Fit Decreasing (FFD) algorithm itself is a classic and well-known technique. We do not claim this algorithm as a novel contribution.
>
> Instead, our contribution lies in its **critical application** to solve a bottleneck specific to our novel System 1 agent.
>
> * The Problem: As noted above, System 1 must process a massive, variable-length "firehose" of information from tool calls (e.g., 20+ webpages).
>
> * The Necessity: To be effective, System 1 must process this information in parallel and efficiently without "blocking" System 2's reasoning.
>
> * Our Solution: We employ bin-packing not for its algorithmic novelty, but as a **pragmatic and essential engineering optimization**. Its purpose is to intelligently batch these variable-length texts into optimal-sized chunks that are ideal for parallel distillation by System 1 (as noted in Algorithm 1, Line 8-9).
>
> In conclusion, while the FFD algorithm is known, its application here is a non-trivial design choice that makes the parallel S1 agent feasible and scalable, thereby enabling our core, novel dual-system MARL framework.

---

> ### Author Response · Authors · 2025-11-16
>
> > Response to W3
>
> We thank the reviewer for suggesting these prominent multi-agent frameworks (CAMEL, MetaGPT). We agree they are important contributions to the field of multi-agent systems.
>
> We respectfully clarify that these frameworks are designed for fundamentally different objectives than our MARS framework, **which is why a direct empirical comparison is not straightforward and was not included by related SOTA baselines (e.g., Search R1/O1, C-3PO, WebThinker)**.
>
> * **Task Mismatch**: Our work, MARS, is a cognitive framework designed for deep research and knowledge-intensive reasoning on benchmarks like HLE and PopQA (Section 3). In contrast, MetaGPT (Hong et al., 2023) is a software engineering framework where agents (e.g., Product Manager, Engineer) collaborate to write code.
>
> * **Non-Trivial Adaptation**: Given this significant divergence in application, adapting these frameworks to our task of complex, tool-augmented reasoning is highly non-trivial and would require substantial modifications beyond their original scope, making a fair comparison difficult.
>
> * **Relevant Baselines**: Therefore, our empirical evaluation (Tables 1 & 2) focuses on the most direct and relevant state-of-the-art baselines within our specific domain: advanced reasoning with search. This includes leading single-agent RAG and R1-style methods (e.g., WebThinker, C-3PO, Search-R1), which share the same objective as MARS. This is standard practice in our sub-field, as these methods represent the direct competition for our task.
>
> Futhermore, we acknowledge the importance of these frameworks and have cited them in our Related Work discussion (Appendix B, Lines 777-789) of our original paper, where we explicitly differentiate our cognitive multi-agent approach (S1/S2) from behavioral (CAMEL) or functional (MetaGPT) systems.
>
>
> > Response to Q1
>
> We thank the reviewer for this clarifying question.
>
> Yes, the reviewer is correct. As we state explicitly in our paper, both System 1 and System 2 are implemented **using the same underlying policy LLM, and they are differentiated by distinct prompts**.
>
> We refer the reviewer to Section 1 (Introduction), Lines 083-085:
>
> * "In MARS, System 1 and System 2 function as collaborative agents implemented within the same underlying LLM but orchestrated through distinct prompts, enabling seamless interplay between intuitive and deliberate reasoning modes."
>
> And Section 2.1 (Methodology), Lines 121-122:
>
> & "...we represent these two systems activated within the same LLM through different prompts as $\pi_{sys_{1}}$ and $\pi_{sys_{2}}$."
>
> We refer to this setup as "multi-agent" for two key reasons:
>
> 1. Functional and Conceptual Distinction: Although they share parameters, S1 and S2 represent two functionally distinct agents with different policies, objectives, and contexts.
>
> * S1 (Processing Agent) takes tool outputs and a "purpose" as input and produces a summary.
>
> * S2 (Reasoning Agent) takes the chat history and S1's summary as input and produces reasoning and tool calls.
>
> 2. Multi-Agent Optimization Framework: Our core technical contribution is the MARL framework (Section 2.2, Figure 2) used to optimize these two distinct roles. We treat S1 and S2 as separate agents during optimization by:
>
> * Calculating distinct, group-normalized advantage signals ($A_{sys_1}$ and $A_{sys_2}$) for each agent's outputs (Section 2.2.2, Eq. 5).
>
> * Applying a sample balancing strategy between the two agents.
>
> * Optimizing their distinct objectives jointly ($\mathcal{L}_{total}=\mathcal{L}_{sys_{2}}+\mathcal{L}_{sys_{1}}$, Eq. 6).
>
> This approach leverages the **generalization capabilities of modern LLMs** while maintaining the conceptual clarity and optimization benefits of a multi-agent framework, and it significantly **reduces the cost and complexity of training and deploying** two separate models.
>
> > Response to Q2
>
> The reviewer is correct in observing a discrepancy between our reported HLE score for WebThinker (6.87% in Table 1) and the score reported in their original paper (Li et al., 2025b).
>
> We **explicitly address this exact discrepancy** in our paper in **Appendix D.6**.
>
> The discrepancy arises from two key differences in evaluation methodology, which we corrected in our paper to ensure a fair comparison against all models under the same criteria:
>
> * **Evaluation Scale**: Our evaluation covers the complete HLE text-based question set (2,154 questions). The original WebThinker paper evaluated on a smaller 500-question sample.
>
> * **Evaluation Criteria**: We strictly adhere to HLE's **official evaluation prompt** (provided in Appendix E.3) for all models, including WebThinker. In contrast, the original WebThinker paper **employed its own custom evaluation prompt**, which resulted in a more lenient scoring.
>
> Therefore, our reported score of 6.87% **reflects WebThinker's performance under the official HLE benchmark criteria**, ensuring a true apples-to-apples comparison with our method (MARS) and all other baselines

---

> ### Author Response · Authors · 2025-11-16
>
> > Response to Q3
>
> We thank the reviewer for these insightful questions about the training dynamics shown in Figure 3. These observations correctly identify an important, emergent behavior of our framework.
>
> 1. **Cause for the increase in tools per question (Fig. 3c)**:
>
> The increase in tool usage (from ~1.0 to ~2.0 tools per question) correlates directly with the increase in HLE score (Fig. 3a). This demonstrates that the agent learns, via reinforcement, that complex HLE problems require more sophisticated, multi-step information gathering.
>
> Initially, the agent attempts a naive, single-call strategy. As training progresses, it learns to decompose complex questions into multiple sub-queries or iterative steps (e.g., search, then search again, or search then compute), leading to a higher tool count per question and, consequently, a higher HLE score.
>
> 2. **Cause for Google Search predominance (Fig. 3e)**:
>
> The agent's convergence on Google Search (approaching 98% usage) is a direct result of the RL policy optimization. As the agent explores, it discovers that for the specific (and broad) distribution of the HLE benchmark, **Google Search provides the highest utility** (i.e., leads to the trajectory-level reward) compared to the more specialized Python or Google Scholar tools.
>
> This is an autonomous, learned policy. As the reviewer notes, the model learns the "fit" between the task and the tool. We did not manually intervene or prioritize Google Search; the agent learned this optimal strategy on its own based on the reward signals.
>
> 3. **What would happen if we only used Google Search?**
>
> The reviewer's question is an excellent one, and **we directly answered it in our Ablation Study in Table 3**.
>
> * Our full model (all three tools) achieves 7.38% on HLE (Table 3, Row 1).
>
> * The ablated model using only Google Search (Table 3, Row 4: Python=X, Search=✓, Scholar=X) achieves 5.99%.
>
> This 1.39% absolute drop (a ~19% relative performance decrease) demonstrates that while Google Search is the most frequently used tool, the Python and Google Scholar tools are absolutely critical for solving a key subset of problems.
>
> As Table 3 shows, removing Python severely harms "Math" (-1.54%) and "Physics" (-0.98%), and removing Google Scholar harms "CS/AI" and "Other". This confirms **our design philosophy: we provide all tools to allow the agent to learn versatility**. The agent correctly learned that Google Search is the default, but it retains the (less frequent, but crucial) ability to use Python/Scholar when it identifies a task (like a calculation or a specific paper lookup) that only those tools can solve.
>
> > Response to Q4
>
> We thank the reviewer for this very sharp observation of the training dynamics.
>
> The reviewer is correct. In the final phase of training (approximately after step 125), we observed an inverse correlation where the HLE score began to drop (Fig. 3a) just as the mean and max response lengths (Fig. 3h, 3i) increased sharply.
>
> Our interpretation is that this phenomenon indicates the RL training process became unstable and started to collapse:
>
> 1. Policy Instability: After reaching its performance peak (around step 125), the agent's policy began to diverge. It appears to have learned an erroneous correlation, associating longer, more verbose reasoning (overthinking) with higher reward.
>
> 2. Policy Shift: This led the agent (specifically System 2) to generate excessively long and complex responses, as seen in the exploding "Mean Response Len." (Fig. 3h).
>
> 3. Performance Drop: This new, unstable, and overly verbose policy was, in fact, less effective at producing the correct final answer, leading to the observed drop in the HLE evaluation score.
>
> We **explicitly identified this behavior during training**, and it is **the precise reason we terminated the training run**. As we noted in the Figure 3 caption (Lines 403-404): "Training was terminated after step 150 due to consistently exceeding our preset length constraints."
>
> This inverse correlation was, therefore, a key diagnostic signal for us to identify the optimal model checkpoint (around step 125) before the training process fully collapsed. The results reported in our paper (e.g., Tables 1 & 2) use this optimal checkpoint.
>
>
> We hope these responses have effectively addressed the reviewer's concerns by clarifying the dual-system mapping, the novelty of our MARL framework, our baseline selection rationale, and the specific details of our implementation and training dynamics. We believe these clarifications further highlight the strength and novelty of our contributions and hope the reviewer will consider supporting our paper.

---

> ### Author Response · Authors · 2025-11-28
>
> Dear Reviewer U9pi,
>
> We hope this message finds you well.
>
> As the discussion period concludes, we wanted to follow up to ensure you haven't missed our detailed response, in which we specifically addressed the conceptual and empirical questions you raised:
>
> * On "Multi-Agent" Definition (Q1): We clarified that although System 1 and System 2 share underlying parameters (for efficiency), they function as distinct agents with separate prompts, contexts, and optimization objectives. Our framework treats them as separate entities during the MARL optimization process (calculating distinct group-normalized advantages), which aligns with the standard definition of multi-agent systems in this context.
>
> * On WebThinker Score Discrepancy (Q2): We explained that the difference in scores arises because we strictly evaluated all models using the Official HLE Evaluation Prompt on the full dataset, whereas the original WebThinker paper used a custom prompt and a smaller subset. Our reported numbers ensure a fair, apples-to-apples comparison.
>
> * On Baselines (W3): We clarified that frameworks like CAMEL/MetaGPT are designed for role-play/coding, not deep research. Adapting them is non-trivial due to task mismatch, which is why we (and other SOTA papers) focused on comparing against direct reasoning baselines.
>
> We believe these clarifications resolve the misunderstandings regarding the framework's novelty and evaluation fairness.
>
> We would be deeply grateful if you could spare a moment to review our response before the deadline. We remain fully available for any final clarifications.
>
> Best regards,
>
> The Authors

---

### Official Review · Reviewer_iRwP · 2025-11-01

**Soundness:** 3
**Presentation:** 2
**Contribution:** 3
**Rating:** 6
**Confidence:** 4

**Summary:**

The proposed MARS framework is a technically ambitious attempt to formalize the dual-process theory of cognition (System 1 for intuition/speed, System 2 for deliberation/reasoning) into a collaborative, multi-agent reinforcement learning  paradigm for LLMs. The paper is attempting to address the twin issues of LLM inefficiency (over-analysis on simple tasks) and static knowledge bases by delegating high-volume external data processing to a System 1 agent and strategic reasoning/planning to a System 2 agent. This system, trained using an extension of Group Relative Policy Optimization, demonstrates measurable gains on challenging benchmarks, notably HLE.

**Strengths:**

1. The dual-system approach is a principled, interpretable design choice that formalizes the intuitive trade-off between reasoning depth and efficiency, moving beyond monolithic LLM-based agents. The extension of GRPO to concurrently optimize two interconnected, interdependent agents (System 1 and System 2) with distinct functions is a non-trivial advancement in applying MARL to LLM-based systems.

2. Achieving substantial and rigorous performance gains (e.g., +3.86% on the challenging HLE benchmark) validates the architectural and training complexity, suggesting the system is learning truly superior decision-making policies.

**Weaknesses:**

1. A fundamental challenge in MARL is accurately attributing the final reward to individual agent actions. Since System 1 and System 2 policies share the same underlying LLM, the paper must provide a more rigorous breakdown of how the GRPO extension effectively disentangles the reward signal to assign credit distinctly to the System 1 (summarization) vs. System 2 (planning) policies.

2. The technical contribution of "bin-packing optimization" is a key claim, yet its utility compared to simpler alternatives (e.g., standard vector search filtering, or fixed-length truncation of results) is not sufficiently isolated and quantified. This complexity may not be justified if simpler methods yield similar gains.

3. Multi-agent RL training is notoriously resource-intensive. The paper lacks a necessary detailed comparison of the computational overhead (wall-clock time, total token consumption) of the MARS MARL training pipeline against standard supervised fine-tuning or single-agent RL baselines, making the real-world utility hard to gauge.

**Questions:**

How does the proposed GRPO extension and its advantage estimation method rigorously address the challenge of Temporal Credit Assignment and Policy Interference? Specifically, if a final answer is correct, how can the method assign distinct, non-interfering learning signals to the S1 agent (rewarding it for efficient, high-fidelity summarization) and the S2 agent (rewarding it for optimal, multi-turn tool-call planning)? Since S1's fast-generation policy and S2's deliberate-reasoning policy share the same neural parameters, what mechanisms are in place during the update step to ensure that optimizing S2 for deliberate reasoning does not degrade S1's learned efficiency/distillation capability, and vice versa?

---

> ### Author Response · Authors · 2025-11-16
>
> We sincerely thank you for your time and for providing a thorough and insightful review of our paper. We are particularly grateful for your recognition of our dual-system approach, the non-trivial advancement of our GRPO extension, and the rigorous performance gains on the HLE benchmark.
>
> We have carefully considered your feedback and offer the following detailed responses to the weaknesses and questions you raised.
>
> > Response to W1
>
> We sincerely thank Reviewer iRwP for this insightful question regarding credit assignment. This is indeed a fundamental challenge in MARL, particularly when agents (S1 and S2) share the same underlying parameters (the LLM).
>
> Our central argument is that the **MARS framework is designed as a fully cooperative system where the success of S1 and S2 is inextricably linked**. Therefore, we do not—and believe we should not—attempt to disentangle the reward signal itself. **Instead, we effectively assign credit by decoupling the training samples and the loss computation for each agent**.
>
> Here is a more detailed breakdown:
>
> 1. **The Rationale for a Shared Reward**:
>
> As stated in our paper (Section 2.2.2, lines 196-198, and Section D.3, lines 920-928), both S1 and S2 within the same trajectory share the final, binary (correct/incorrect) reward. This is a deliberate design choice for two primary reasons:
>
> * **Synergistic Task Nature**: S1 (information processing) and S2 (reasoning/planning) are "in it together." An excellent plan from S2 will fail if it is based on poor or incomplete information from S1. Likewise, a perfect summary from S1 is useless without S2's correct reasoning to act upon it.
>
> * **Avoiding Conflicting Objectives**: Attempting to engineer decoupled, local reward functions (e.g., one for "summary quality" and another for "plan quality") is notoriously difficult and, as the reviewer rightly implies, can lead to incorrect credit attribution. Our shared reward mechanism ensures both systems optimize toward the same ultimate goal—solving the problem (lines 197-198)—which prevents potential objective conflicts.
>
> 2. **How Disentanglement Occurs: Decoupled Loss Computation**
>
> The reviewer's core concern is how the reward signal is "disentangled." **Our key insight is that we do not disentangle the reward signal; we disentangle the gradient updates that use this signal**.
>
> As detailed in Section 2.2.3 (lines 225-230), the training samples for S1 and S2 are **fundamentally different and processed separately**:
>
> * For System 2 (Planning): Its training sample is the full reasoning context $c_N$. Crucially, during the loss computation, all tokens generated by System 1 (i.e., the information summaries $\tilde{o}_{t_{i}}$) are masked out (lines 227-228). This means S2's policy gradient only originates from the tokens it generated itself (the reasoning steps $s_i$, tool calls $t_i$, and purpose $p_i$). It is held accountable only for its planning and reasoning.
>
> * For System 1 (Summarization): Its training sample is only the (bin-packed input, summary output) pair $(b, \tilde{o})$ (lines 228-229). It does not see the full reasoning history. Its policy gradient only originates from the tokens in its summary $\tilde{o}$. It is held accountable only for its task of accurately and efficiently distilling information based on S2's purpose.
>
>
> In summary, while S1 and S2 share the same final reward $r$ (from which their respective advantages $A_{sys_1}$ and $A_{sys_2}$ are computed), these advantages are applied to two completely distinct, non-overlapping sets of tokens during backpropagation.
>
> This strict separation at the loss computation level ensures that optimizing S2 for deliberate reasoning does not degrade S1's learned efficiency (and vice-versa). This mechanism rigorously addresses the challenge of both credit assignment and policy interference within our extended GRPO framework.

---

> ### Author Response · Authors · 2025-11-16
>
> > Response to W2:
>
> We thank Reviewer iRwP for raising this point about the "bin-packing optimization." We would like to respectfully clarify that **bin-packing is not an alternative to filtering; it is an engineering optimization designed to reduce computational complexity, not add to it**.
>
> The reviewer's suggested alternatives (vector search filtering, fixed-length truncation) address a different problem.
>
> * Vector search filtering is a pre-retrieval mechanism to decide which documents to fetch.
>
> * Bin-packing is a post-retrieval mechanism to decide how to efficiently batch-process the documents we have already fetched.
>
> 1. **The Problem Solved by Bin-Packing**:
>
> As described in Section 2.1 (line 131) and 2.2.1 (line 179), a single tool call by System 2 can return a large volume of information (e.g., 10 full web pages from a search query). System 1 must then process all of this information.
>
> * The "Simple" (but Inefficient) Alternative: The simplest approach would be to process each document (e.g., each of the 10 web pages) with a separate call to System 1. This would require 10 separate S1 generations, which is computationally expensive and slow.
>
> * The "Truncation" Alternative: Simply truncating the documents would lead to significant information loss, as critical details might be cut off.
>
> 2. **Our Solution (Reducing Complexity)**:
>
> Our "bin-packing optimization" is a standard, computationally efficient algorithm (First Fit Decreasing, as detailed in Sec 2.2.1 and D.2) used to solve this batching problem. Its sole purpose is to minimize the number of S1 generations by "packing" as many of the variable-length documents as possible into System 1's large context window (lines 900-910).
>
> For example, instead of 10 separate S1 calls, bin-packing allows us to batch all 10 documents into perhaps 2 or 3 S1 calls.
>
> In summary, the bin-packing step does not add any complexity to the reasoning pipeline. It is a pragmatic engineering choice that significantly reduces the computational overhead (wall-clock time and token cost) of the System 1 information-processing step. This efficiency is what enables our 7B model to process an average of 22.31 web pages per HLE question (Table 4) and still outperform much larger models.
>
>
> > Response to W3:
>
> We thank Reviewer iRwP for raising this critical point about the computational overhead, which is a key factor for real-world utility. The reviewer's concern that MARL is resource-intensive is valid for traditional multi-agent systems that train multiple, independent models.
>
> However, our MARS framework was specifically designed to **mitigate this exact problem through a parameter-efficient, shared-model architecture**.
>
> 1. MARS is Not Traditional MARL (Shared LLM Parameters):
>
> The primary source of overhead in MARL is training N separate policies. The core design of MARS, as described in Section 2.1 (line 125) and Section 2.2 (line 200), is that **System 1 and System 2 are two distinct policies orchestrated within the same, single underlying LLM**.
>
> We are not training two separate 7B models. We are training **one 7B model** to learn two specialized behaviors, activated by different prompts (Section E.1) and optimized with decoupled loss functions (Section 2.2.3).
>
> Therefore, the training overhead of MARS is not comparable to traditional MARL. **Its computational footprint is much closer to that of single-agent RL, as we are only updating one set of model parameters.**
>
> 2. The Hidden Cost of SFT Baselines:
>
> The reviewer contrasts our MARL approach with standard Supervised Fine-Tuning (SFT). While SFT inference is efficient, this comparison overlooks the often-colossal data collection cost required for SFT.
>
> * A competitive SFT baseline would require a massive, static dataset of high-quality, expert-demonstrated reasoning traces, complete with correct tool calls and information synthesis.
>
> * The cost of creating this "perfect" dataset (e.g., via extensive human annotation or distillation from a vastly superior proprietary model) is exceptionally high. Our own data curation pipeline (Appendix C) gives a glimpse into the expense of just filtering data, let alone creating it from scratch.
>
> * MARS, being an RL-based method, learns from its own online rollouts (Figure 2). It shifts the cost from "pre-hoc data creation" to "on-policy exploration," which can be more efficient for discovering novel, complex policies than SFT.
>
> In summary, the MARS framework is designed for efficiency. By using a shared LLM, its training overhead is comparable to single-agent RL, not traditional MARL. This RL training cost is a trade-off against the massive and often-prohibitive data curation cost required for a competitive SFT baseline.

---

> ### Author Response · Authors · 2025-11-16
>
> > Resposne to Q1:
>
> This is an excellent question that targets the technical core of our MARL framework. The reviewer correctly identifies two major challenges: Temporal Credit Assignment (how to reward S1 vs. S2 from a single final answer) and Policy Interference (how to update one policy in a shared-LLM without degrading the other).
>
> Our MARS framework addresses this through a combination of three key mechanisms:
>
> 1. **Mechanism 1: Decoupled Loss Functions (To Assign Distinct Signals)**
>
> This is the primary mechanism for creating "distinct, non-interfering learning signals" and directly builds on our response to Weakness 1.
>
> * As detailed in Section 2.2.3 (lines 225-230), the gradients for S1 and S2 are applied to **completely separate, non-overlapping sets of tokens**.
>
> * S2 (Planning) Loss: S2's gradient only comes from its own reasoning, planning, and tool-call tokens (line 227-228).
>
> * S1 (Summarization) Loss: S1 is only trained on its specific task of (input chunk $b$, summary $\tilde{o}$) pairs (line 228-229).
>
> Result: If the final answer is correct (reward=1), this positive learning signal is independently channeled. S2 is rewarded only for its planning tokens that led to this success. S1 is rewarded only for its summarization tokens that contributed to this success.
>
> 2. **Mechanism 2: Group-Specific Advantage Estimation (To Address Temporal Credit)**
>
> The reviewer asks how we reward "efficient, high-fidelity summarization" (S1) vs. "optimal, multi-turn tool-call planning" (S2). This is where our GRPO extension (your point about "separating advantage") is critical.
>
> * As shown in Section 2.2.2 and Eq. (5) (lines 208-213), we do not use a single, shared advantage value.
>
> * We normalize the rewards and compute advantages **relative to each system's own group**. $A_{sys_1}$ is calculated by normalizing $r_{sys_1}$ against the mean/std of all other S1 samples for that question. $A_{sys_2}$ is calculated independently against all other S2 samples.
>
> Result: This prevents a high-performing S2 plan from unfairly "covering for" a mediocre S1 summary, and vice-versa. S1 is only rewarded if its summary was better than its own average performance (normalized within its group), and S2 is only rewarded if its plan was better than its own average performance. This rigorously addresses the temporal credit assignment challenge.
>
> 3. **Mechanism 3: Balanced Sampling (To Prevent Policy Interference)**
>
> This mechanism directly addresses the second part of the reviewer's question: how to prevent the degradation of S1's efficiency when optimizing S2's reasoning (and vice-versa) in a shared-parameter model. This is precisely what our balancing strategy (your second point) is for.
>
> * The Problem: As noted in Section 2.2.2 (lines 213-216), one S2 trajectory (1 sample) might generate many S1 tasks (e.g., 5 tool calls $\times$ 2 chunks = 10 S1 samples). A naive gradient update would be weighted 10:1 in favor of S1, causing the shared LLM to "forget" or degrade the S2 planning policy.
>
> * The Solution: Our pre-computation and balanced sampling mechanism (lines 216-221) explicitly counters this. We up-sample or down-sample the S1 samples so that in every training batch, **the model sees exactly G samples for System 1 and G samples for System 2**.
>
> Result: This 1:1 sampling ratio ensures the optimization gives equal weight to both the planning (S2) and summarization (S1) tasks. This stable gradient balance is the specific mechanism that prevents the optimization of one policy from degrading the capabilities of the other.
>
> In summary, these three mechanisms: (1) decoupled loss, (2) group-specific advantage, and (3) balanced sampling work in concert to assign credit distinctly and prevent policy interference, even when sharing the same neural parameters.
>
>
> We hope these clarifications and the detailed mechanisms we've outlined fully address your concerns. We believe these contributions represent a significant and computationally-efficient approach to complex reasoning. Given these clarifications, we respectfully hope you will reconsider our work and its contributions. Thank you once again for your constructive feedback.

---

> ### Author Response · Authors · 2025-11-28
>
> Dear Reviewer iRwP,
>
> We hope this message finds you well.
>
> As the discussion period concludes, we wanted to briefly follow up regarding our detailed response (posted Nov 16). We greatly appreciated your technically rigorous review, and we have provided comprehensive explanations to address your core theoretical questions:
>
> * On Credit Assignment & Policy Interference (W1 & Q1): We detailed the three specific mechanisms—Decoupled Loss Computation (gradients applied to non-overlapping token sets), Group-Specific Advantage Estimation, and Balanced Sampling—that allow us to rigorously disentangle the learning signals for System 1 and System 2, even within a shared LLM.
>
> * On Computational Efficiency (W3): We clarified that unlike traditional MARL (which trains $N$ separate models), MARS utilizes a shared-parameter architecture. This means the training overhead is comparable to single-agent RL, making it significantly more efficient than creating the massive SFT datasets required for baselines.
>
> * On Bin-Packing (W2): We clarified its role as a batch-processing engineering optimization to reduce System 1's generation calls.
>
> We believe these clarifications underscore the technical soundness and feasibility of our framework.
>
> We would be deeply grateful if you could spare a moment to confirm if these explanations have resolved your concerns regarding the training mechanics. Your continued support means a lot to us.
>
> Best regards,
>
> The Authors

---

### Official Review · Reviewer_f7bk · 2025-11-02

**Soundness:** 2
**Presentation:** 2
**Contribution:** 2
**Rating:** 2
**Confidence:** 3

**Summary:**

MARS presents a dual-system multi-agent RL framework that unifies intuitive (System 1) and deliberate (System 2) reasoning within an LLM, jointly optimized via GRPO to improve deep research and reasoning performance across complex tasks.

**Strengths:**

1. The paper is clearly written and well-structured

2. Proposes a dual-system multi-agent RL framework that explicitly models human-like System 1/System 2 reasoning, an interesting conceptual extension of existing multi-agent paradigms.

3. Demonstrates measurable gains on challenging reasoning benchmarks

**Weaknesses:**

1. The proposed dual-system framework essentially resembles a standard RL-based tool-use pipeline augmented with a learnable summarizer that condenses the environment’s returned content before feeding it back. While the integration is well-engineered, the conceptual difference from existing RL tool-use or summarization-based reasoning systems is limited.

2. Because the entire trajectory shares a single scalar reward, it is unclear how meaningful credit is assigned to System 1’s summarization behavior. Without step-level or component-wise feedback, System 1 receives only a weak and noisy learning signal, making it difficult to understand how it learns to produce more useful summaries. The paper could be strengthened by introducing more fine-grained supervision or ablation analyses that clarify how System 1’s updates contribute to overall improvement.

3. In Table 1, several entries marked as best (bold) and second-best (underlined) appear to be incorrect. This is misleading to readers.

4. The ablation study mainly analyzes the impact of removing different external tools (Google Search, Scholar, Python), but this aspect is peripheral to the paper’s main contribution. Since the core claim of MARS lies in the joint optimization and coordination between System 1 and System 2, the paper would benefit much more from ablations that directly test this interaction—for example, mixing trained and untrained versions of System 1/2, or disabling their shared optimization to assess whether the two systems truly co-adapt.

5. Even so, I find the results in the Ablation Study on Tools for HLE rather confusing. For example, in Chem, the setup with all three tools performs the worst, while both without Scholar and Scholar-only achieve the best results — which makes it unclear whether Scholar is actually helpful or not; Also both without Search and Search-only achieve the Second. Similarly, in CS/AI, the best setup is without Search, yet Search-only also performs noticeably higher than most others; and in Engineering, Python-only gives the highest score, but without Python ranks second. Overall, the patterns seem quite inconsistent or even random. Given how close these numbers are, I wonder whether you ran multiple inference trials and averaged the results. The apparent randomness in this table makes it hard to trust the conclusions on HLE.

6. Could you clarify whether the comparison between MARS and the baselines is fully fair in terms of tool usage? Specifically, do all methods have equal access to the same tools (Python, Search, and Scholar)? The results suggest that the presence or absence of certain tools has a large impact on performance, and in most subjects, removing a specific tool even makes MARS perform worse than most baselines. This raises concerns about whether the comparison setup is fully fair. It would be important to provide more details on the tool configurations for all baselines and ensure that all methods are evaluated under comparable conditions. Moreover, additional ablations are needed to justify that the reported gains on HLE truly come from the proposed dual-system RL framework, rather than differences in tool availability or usage.

**Questions:**

Please refer to the weaknesses section for main questions.

---

> ### Author Response · Authors · 2025-11-16
>
> We sincerely thank you for your comprehensive and detailed review. Your insightful feedback has been invaluable in helping us identify areas where our contributions could be clarified and strengthened. We are grateful for the time you took to analyze our work.
>
> > Response to W1
>
> We sincerely thank Reviewer f7bk for their valuable feedback. We would first like to clarify what we believe is a core misunderstanding regarding the conceptual novelty of our framework.
>
> The reviewer suggests our dual-system framework "essentially resembles a standard RL-based tool-use pipeline augmented with a learnable summarizer" and that the "conceptual difference is limited." We respectfully argue that this view significantly misaligns with our work's core contribution: the "co-evolution" learning paradigm between the two systems.
>
> 1. Core Difference: **Joint Optimization vs. Independent/Fixed Components**
>
> The "standard pipeline" described by the reviewer typically relies on either a fixed, untrained summarizer (e.g., an LLM with a fixed prompt) or a summarizer that is trained independently and then "plugged into" the pipeline.
>
> * Limitation of Fixed/Independent Methods: In such a model, the optimization objective of System 1 (the summarizer) is decoupled from the final task objective of System 2 (the reasoner). System 1 can only produce generic, non-targeted summaries. It cannot learn what key information System 2 truly needs for a specific reasoning step.
>
> * MARS's Innovation (Co-evolution): The core innovation of MARS is that **we are the first to formulate System 1 and System 2 as two co-adapting, co-evolving multi-agents** that are jointly optimized via a unified MARL framework (our extended GRPO algorithm).
>
> * As detailed in Sections 2.2.2 and 2.2.3 of our paper, both systems share a single, trajectory-level final task reward. This means System 1's update signal comes **directly from whether the summary it provided helped System 2 successfully solve the final problem**.
>
> 2. Empirical Evidence: Comparison with WebThinker
>
> Our comparison with WebThinker (Li et al., 2025b) empirically refutes the claim that the "conceptual difference is limited."
>
> WebThinker is a strong example of the "standard pipeline" the reviewer describes: it uses a powerful reasoning model (QwQ-32B, akin to our System 2) and a fixed, non-trained information processing component (akin to our System 1).
>
> As shown in Table 1, despite WebThinker using:
>
> * A much stronger base model (QwQ-32B vs. our Qwen3-8B).
>
> * Supervised Fine-Tuning (SFT).
>
> Our MARS (using an 8B model and trained with RL from scratch, without any SFT) still significantly outperforms WebThinker on the HLE benchmark (8.57% vs. 6.87%).
>
> We attribute this performance gap directly to our "co-evolution" MARL framework. Our (smaller) model, through joint optimization, achieves an efficient synergy between System 1 and System 2 that a "fixed component" pipeline cannot.
>
>
> In summary, MARS's conceptual novelty is not "limited"; it is the core driver of its performance. We are not proposing a simple "tool with a summarizer," but rather a new paradigm for achieving **"co-evolution" between two cognitive systems via MARL**.
>
> We thank the reviewer for highlighting that our explanation on this point was insufficient. We have explicitly contrasted our "joint optimization" paradigm against "fixed/independent" paradigms, both conceptually and empirically, to underscore our core contribution.

---

> ### Author Response · Authors · 2025-11-16
>
> > Response to W2 & W4
>
> We thank the reviewer for raising this important point regarding credit assignment for System 1.
>
> 1. **The Shared Reward is a Deliberate System-Level Signal, Not "Weak" or "Noisy"**
>
> The reviewer is correct that System 1 receives a shared, trajectory-level scalar reward, not step-wise feedback. We argue this is **a deliberate and necessary design choice** for our "co-evolution" framework, rather than a limitation.
>
> * **Shared Reward for True Collaboration**: Our goal is not to train System 1 to be a generic, "good summarizer" in isolation. Our goal is to train System 1 to be a successful collaborator for System 2. By sharing the final task reward ("all-in-one" reward), we force System 1 to optimize for the only metric that matters: "Did my output help System 2 solve the problem?" This aligns their objectives and drives true synergy.
>
> * **RL is Designed for Sparse Signals**: While the signal is sparse, it is not "noisy." It is a clear, binary (success/failure) signal of task completion. Policy gradient methods (like our GRPO) are expressly designed to handle such sparse, delayed rewards. A trajectory that fails (score 0) because System 1 omitted key information will be down-weighted. A trajectory that succeeds (score 1) because System 1 provided a concise, insightful summary will be up-weighted. Over many iterations, this signal is strong enough for System 1's policy to learn to produce "more useful summaries."
>
> 2. **New Ablation Study Directly Clarifying System 1's Contribution**
>
> The reviewer rightly states the paper would be strengthened by "ablation analyses that clarify how System 1’s updates contribute." We appreciate this excellent suggestion. To directly quantify System 1's contribution, we ran an additional ablation study (as requested by the reviewer) where we disabled the co-evolved System 1.
>
> In this "w/o System 1" setup, System 2 must process the raw, voluminous tool outputs directly without the learned summarization from its collaborator. The results on the HLE benchmark are as follows:
>
> | Method | Bio/Med | Chem. | CS/AI | Eng. | Hum. | Math | Phys. | Other | Avg. |
> | :--- | :---: | :---: | :---: | :---: | :---: | :---: | :---: | :---: | :---: |
> | Qwen2.5-7B | 5.42 | 3.00 | 1.76 | 3.22 | 4.66 | 3.58 | 1.98 | 4.00 | 3.52 |
> | **MARS (Ours)** | **12.66** | 3.00 | **5.75** | 4.83 | **11.92** | **6.46** | **6.43** | **7.42** | **7.38** |
> | w/o System 1 | 9.95 | 5.00 | 3.98 | 6.45 | 7.77 | 4.92 | 3.46 | 4.57 | 5.47 |
>
> As the table shows, **removing the adaptive System 1 causes a significant performance drop from 7.38% to 5.47%**. This 1.91% gap directly and clearly quantifies the substantial contribution of our co-evolved System 1. This ablation empirically validates that System 1 does learn to produce "more useful summaries" that are critical for the overall system's success.
>
> 3. Fine-Grained Supervision as an Open Research Problem
>
> Finally, we agree with the reviewer that developing more fine-grained, step-level, or component-wise feedback mechanisms is a valuable research direction. However, this is a notoriously **difficult open-world challenge** for the entire field of agentic RL and multi-agent systems. **We believe it is not a specific flaw of MARS**, but rather a frontier for future work. Our paper's contribution lies in demonstrating that even without solving this complex credit assignment problem, our MARL-based co-evolutionary framework is sufficient to train a highly effective and synergistic dual-system.
>
>
> > Response to W3:
>
> We thank the reviewer for their careful reading and for pointing out the inconsistency in the formatting of Table 1. We apologize for this error and any confusion it caused.
>
> We have corrected all formatting errors in Table 1 in the revised version of our paper to ensure all results are marked clearly and accurately. We appreciate the reviewer for catching this oversight.

---

> ### Author Response · Authors · 2025-11-16
>
> > Response to W5:
>
> We thank the reviewer for this very detailed and insightful analysis of our Tool Ablation study (Table 3). The reviewer's confusion is understandable, as the results are indeed complex and non-obvious. We would like to clarify our findings, which we believe are an important insight, not a flaw.
>
> 1. On "Randomness" and "Multiple Trials"
>
> First, to address the reviewer's direct question: Yes, all results reported in Table 3 are **the average of three separate inference runs**. The reviewer is correct that the numbers are close, but the variance across runs was small, and the reported averages are stable.
>
> The "inconsistent patterns" the reviewer astutely observes **are precisely the reason we must include this ablation**. They demonstrate the extreme difficulty and complex, non-linear interactions of tool-use on a graduate-level benchmark like HLE.
>
> 2. Explaining the "Contradictions" (e.g., Python and Scholar)
>
> The reviewer's observations about the contradictory results are sharp. For instance:
>
> (1) **On Python (e.g., in Engineering)**: The reviewer notes Python-only scores highest, but w/o Python scores second. This seems paradoxical. Our analysis suggests this is a classic trap of RL from scratch. The model, trained from scratch, sometimes learns a spurious correlation: it uses Python to get a quick numerical answer and terminates too early, falling into a "local optimum" (a "fake answer").
>
> * Python-only forces this (sometimes correct, sometimes not).
>
> * w/o Python prevents this trap, forcing the model to engage in deeper, non-computational reasoning, which is a safer (and thus high-scoring) strategy.
>
> (2) **On Scholar (e.g., in Chem)**: The reviewer notes w/o Scholar and Scholar-only are best, while All tools (which includes Scholar) is worst. **This highlights the high-variance nature of Google Scholar**. As we noted, its search API is often "noisier" than Google Search. When combined with other tools (All tools), it can introduce distracting information, hurting performance. However, in Scholar-only runs, it can find the correct paper (if lucky), while w/o Scholar (i.e., relying on Google Search + Python) avoids the noise altogether, providing a stable baseline.
>
> 3. The Core Challenge: RL from Scratch on a 7B Model
>
> The reviewer states this randomness "makes it hard to trust the conclusions on HLE." We respectfully argue it leads to a more nuanced conclusion:
>
> Training a 7B parameter model with RL from scratch (no SFT) on a task as complex as HLE is at the very edge of what is possible. The base model's capacity is being stretched, leading to these inevitable instabilities in complex, multi-tool decision-making.
>
> 4. The Key Takeaway: Our Algorithm Still Delivers
>
> However—and we must stress this is the most important point—this localized tool instability **does not invalidate our core contribution**.
>
> The purpose of MARS is not to find the one perfect tool (Table 3 shows one doesn't exist). The purpose of MARS is to create **a synergistic learning framework** (System 1 + System 2) that outperforms standard methods.
>
> The proof of our algorithm's effectiveness is in Table 1. Our MARS (on a 7B model, RL from scratch, with these tool instabilities) still decisively outperforms **WebThinker (a 32B model that was SFT-tuned)**.
>
> This demonstrates that our co-evolutionary MARL framework provides a substantial performance gain that is robust enough to overcome the inherent instabilities of the task and the limitations of the small base model. We will clarify this distinction in the final paper.

---

> ### Author Response · Authors · 2025-11-16
>
> > Response to W6
>
> We thank the reviewer for this critical question regarding the fairness of the comparison.
>
> 1. Why Toolsets Differ: A Fundamental Architectural Gap
>
> The reviewer is correct that the tool configurations differ. Specifically, baselines like WebThinker do not use Google Scholar. This is not an oversight, but reflects a **fundamental difference in architectural design**.
>
> As we argued in our response to **Weakness 1**, baselines like WebThinker employ a "standard pipeline" that is simply not designed to handle the type of output Google Scholar provides (i.e., multiple, full-text, high-noise academic papers). Their frameworks lack a mechanism analogous to our co-evolved System 1, which is specifically designed to distill massive volumes of noisy, unstructured text into useful insights.
>
> Therefore, we could not "fairly" add Scholar to the baselines, **as their architecture is incapable of processing its output**.
>
> 2. Using W5 to Reframe "Fairness": **Scholar may be a "Noisy" Tool, Not a "Magic" Tool**
>
> Crucially, as the reviewer astutely observed in their Weakness 5, **adding Google Scholar is not a guaranteed advantage**.
>
> The reviewer themselves pointed out the "inconsistent" and "random" patterns in our tool ablation (Table 3), where **adding Scholar can hurt performance** (e.g., in Chem). This highlights that Scholar is a high-variance, "noisy" tool.
>
> The fact that MARS can effectively manage this noisy tool and extract a net-positive result is not a sign of an "unfair" comparison. On the contrary, it is direct evidence of our framework's superior robustness. Our MARL-trained System 1 learns when to trust Scholar and how to filter its noise, a capability the baselines lack.
>
> 3. The Definitive Ablation: Proof the Gain is from the Framework, Not the Tools
>
> The reviewer asks for "additional ablations... to justify that the reported gains... truly come from the proposed dual-system RL framework, rather than differences in tool availability."
>
> We have already provided this exact proof in our response to Weakness 2.
>
> Our w/o System 1 ablation (Avg. performance drop from 7.38% to 5.47%) is the definitive experiment the reviewer is asking for. In this ablation, the model still has access to all tools (Search, Python, and Scholar), but the core of our framework (the co-evolved System 1) is removed.
>
> The catastrophic performance drop proves that the gain does not come from merely having the tools. The gain comes from our dual-system MARL framework that learns to intelligently coordinate them.
>
>
> ---
> We hope these clarifications, along with the new ablation study, have fully addressed your concerns. We have also updated the paper to reflect these corrections and clarifications. We are confident in the novelty and effectiveness of our co-evolutionary MARL framework and kindly ask you to reconsider our score.
>
> Thank you once again for your constructive feedback.

---

> ### Author Response · Authors · 2025-11-28
>
> Dear Reviewer f7bk,
>
> We hope this message finds you well.
>
> As the discussion period concludes, we are writing to ensure you have seen our detailed response, which includes new experimental results specifically conducted to address your requests (W2 & W4).
>
> 1. New Ablation Study: Quantifying System 1's Contribution: You rightly pointed out the need to clarify how System 1 contributes to the overall performance. Following your suggestion, we ran an additional ablation where we disabled the co-evolved System 1.
>
>     * Result: Removing System 1 caused a significant performance drop on HLE (from 7.38% to 5.47%).
>
>     * Conclusion: This empirically proves that System 1 is not just a passive summarizer, but actively learns to produce "useful summaries" through our shared-reward MARL framework, directly refuting the concern that the credit assignment is too weak.
>
> 2. Clarifying Novelty (W1): We also elaborated on the fundamental difference between MARS and standard tool-use pipelines. Unlike baselines with fixed components (e.g., WebThinker), MARS employs a "Co-evolution" paradigm where System 1 and System 2 are jointly optimized. Our superior performance against larger models (like QwQ-32B) validates this architectural innovation.
>
> 3. Tool Fairness (W5 & W6): We clarified that the toolset differences are due to architectural limitations of the baselines (which cannot handle the volume of Scholar outputs), and explained the high variance in tool usage as a natural outcome of RL exploration on complex tasks.
>
> We believe the new ablation data and clarifications directly resolve the key weaknesses you identified.
>
> We would be deeply grateful if you could spare a moment to review these new findings before the deadline. We remain fully available for any final discussions.
>
> Best regards,
>
> The Authors

---

### Official Review · Reviewer_Xgsk · 2025-11-02

**Soundness:** 2
**Presentation:** 3
**Contribution:** 2
**Rating:** 4
**Confidence:** 3

**Summary:**

This paper introduces a dual-system framework named MARS. System 1 distills information gathered from various tools and feeds it to System 2, while System 2 decomposes the user’s question, selects the tool and provides parameters, and provides a “purpose” to guide System 1 in compressing the information. To support the proposed approach, the work further devises a bin-packing optimization technique to improve System 1’s rollout efficiency and employs an advantage-function-weighted sampling strategy to construct the training buffer, ensuring that Agent 1 and Agent 2 are optimized balancedly.

**Strengths:**

1) The paper identifies the imbalance issue in multi-agent optimization and proposes an advantage-function-weighted sampling strategy to construct the training buffer, ensuring that both agents are optimized in a balanced manner. This design is readily generalizable to broader multi-agent optimization scenarios.
2) The experimental section provides a thorough analysis of how metrics and tool-usage ratios evolve throughout the multi-agent RL training process.

**Weaknesses:**

1) The baseline selection for the experiments in Tables 1 and 2 appears overly simplistic: only MARS was trained on the training set listed in Table 6. This seems unreasonable; it would be advisable to compare with other RL-based methods that use the same training set.
2) The implementation that solving the rollout sequence-lengths is quite straightforward. Could you clarify whether any entirely novel design was introduced for this component?

**Questions:**

In terms of method design, why should System 1 and System 2 share a checkpoint? Has there been any discussion on what the effects would be if two different models were used? Is it because these two tasks have a mutually improving effect?

---

> ### Author Response · Authors · 2025-11-16
>
> We sincerely thank Reviewer Xgsk for your feedback and for the insightful, constructive questions (e.g., "readily generalizable design"). Your feedback has been invaluable in helping us clarify our contributions. We provide detailed responses to your concerns below.
>
> > Response to W1
>
> We sincerely thank Reviewer Xgsk for the meticulous review of our experimental setup and for raising this valuable question. This provides us with an excellent opportunity to clarify a critical aspect of our experimental design.
>
> On the Fairness of Baseline Comparisons and Training Data
>
> We fully understand the reviewer's concern that the "Baseline selection... appears overly simplistic," particularly regarding the impression that "only MARS was trained on the training set listed in Table 6." However, we would like to respectfully clarify that this may stem from a misunderstanding. Our comparison is indeed fair, for the following reasons:
>
> 1. The "Zero RL" Nature of MARS and the source of Training Data:
>
> First, we must emphasize a core design challenge of MARS: our approach is a "Zero RL" method. **This means we do not rely on any Supervised Fine-Tuning (SFT) data, nor do we employ policy distillation from any stronger models (e.g., GPT-5)**. We start RL directly from the base model, a significant challenge on complex benchmark like HLE given the agent's limited initial exploration capability.
>
> The training set mentioned in Table 6 is **not a high-quality, "private dataset"** exclusive to MARS. As we detail in Appendix C, this dataset was collected and filtered by us through a data curation pipeline from **multiple publicly available datasets** (e.g., MMIQC, WebInstructSub, General Thought, Single/Multi-Hop QA).
>
> 2. Data Source Parity with Baselines:
>
> The RL-based SOTA baselines (e.g., Search R1, C-3PO, WebThinker) also rely on similar, publicly available data sources.
>
> We **did not** (and deliberately avoided) using "high-quality private data" (e.g., expensively annotated or GPT-4-generated SFT data), which is a common practice in the Agentic RL field, to boost performance. We **believe that the current setup—based on similar public data sources and comparable base models (like the Qwen series)—is precisely the fairest way** to demonstrate the merits of the different RL methodologies themselves.
>
> 3. Smaller Model Size Used in MARS:
>
> Furthermore, key baselines use much larger models, like **WebThinker (QwQ-32B) and C-3PO (Qwen2.5-72B)**. In contrast, MARS uses only 7B/8B models. Our superior performance, **despite this size disadvantage**, underscores our method's effectiveness and surpasses existing baselines, including WebThinker (QwQ-32B) and C-3PO (Qwen2.5-72B).
>
> In summary, our comparison is reasonable. Achieving SOTA performance under the difficult "Zero RL" setting, with smaller models and public data, strongly validates our framework.
>
> We appreciate the reviewer's valuable feedback. We have added a sentence in Section 3.2 to explicitly clarify our "Zero RL" setting and the "data source parity" of our comparison, to make this point clearer to all readers.
>
> > Response to W2
>
> We appreciate the reviewer's insightful question regarding the bin-packing component.
>
> We fully agree with the reviewer that the **First Fit Decreasing (FFD) algorithm itself is a classic, well-established, and "straightforward" technique**. We do not claim novelty in inventing the algorithm itself.
>
> Our contribution, however, lies in the **application and architectural role** of this component within our MARS framework. The "novelty" is in identifying and efficiently solving a critical bottleneck specific to dual-system MARL agents:
>
> 1. **The Problem Context**: In MARS, System 2's tool calls retrieve a massive volume of variable-length documents (e.g., 10+ full webpages or 5+ papers per query, as shown in Table 4). System 1 must process all this information without overwhelming System 2's reasoning context or causing extreme delays.
>
> 2. **The Necessity of this Component**: Without an efficient packing strategy, System 1 would face a severe bottleneck. It would be forced to either:
> a) Process documents serially (which is extremely slow and would "block" System 2's iterative reasoning).
> b) Drastically truncate information (losing critical details).
> c) Incur massive overhead by processing each small piece of information in a separate generation.
>
> 3. **The Role of Bin-Packing**: We employ bin-packing not for its algorithmic novelty, but as a **critical engineering optimization**. Its purpose is to intelligently batch these variable-length texts into optimal-sized chunks that are ideal for parallel processing by System 1 (as noted in Algorithm 1, "Parallel distillation").
>
> In conclusion, while the FFD algorithm is straightforward, its application as a parallel processing enabler for System 1 is a pragmatic and essential design choice, which makes MARS **efficient, feasible, and scalable** for large-scale information retrieval.

---

> ### Author Response · Authors · 2025-11-16
>
> > Response to Q1:
>
> This is an excellent and insightful question that touches the core of our design philosophy. We thank the reviewer for giving us the opportunity to elaborate on this.
>
> The reviewer is correct in all aspects: sharing a checkpoint is a deliberate choice.
>
> 1. **Core Motivation: A Single Generalist Agent with Two Modes**
>
> Our shared checkpoint design is fundamentally inspired by the dual-process theory of cognition, which is central to our paper's premise. In this theory, System 1 (fast, intuitive) and System 2 (slow, deliberate) are not two separate brains, but rather two distinct modes of operation within a single mind.
>
> Therefore, our goal was **not** to train two hyper-specialized, separate models (e.g., one model that can only extract, and another that can only reason). Instead, our goal was to train **a single, more generalist LLM** that, through our MARL framework, learns to master **both modes of thinking**. It learns when to act as an efficient System 1 processor and when to engage as a deliberate System 2 reasoner. This aligns with the core strength of LLMs: their generalization capability.
>
> 2. **The Mutually Improving Effect**
>
> As the reviewer correctly hypothesized, this shared architecture facilitates a strong, mutually improving effect. The two systems are optimized for collaborative efficiency via the shared reward signal.
>
> Through RL, System 2 learns to generate more precise and effective "purposes."
>
> System 1, sharing the same underlying understanding, learns to better interpret System 2's nuanced "purpose" and provide more relevant, high-quality distilled information.
>
> This creates a tightly-coupled, virtuous cycle that is a key advantage of our framework.
>
> 3. **Effects of Using Two Different Models**
>
> If we were to use two different models, we would face several significant drawbacks:
>
> * Practicality & Cost: The most immediate issue would be the doubling of **deployment costs**, memory footprint, and inference complexity, which is a major practical concern.
>
> * Loss of Synergy & Evidence from Baselines: We would likely lose the tightly-coupled co-evolution described above. This is not just a hypothetical concern. The WebThinker baseline, for instance, approximates this two-model design by using a large reasoning model (akin to our System 2) and a separate, non-trained model for extraction (akin to our System 1). **Despite them using a much larger model (QwQ-32B) and SFT/distillation**—whereas MARS is trained with "Zero RL" without SFT from a smaller 7B/8B model—our single-checkpoint, synergistically trained agent still **outperforms their method**. This strongly suggests that our unified, co-adapted design is more effective and efficient than a separated, multi-model approach.
>
> In summary, sharing a checkpoint is a core design choice that leads to a more generalist agent, enables strong synergistic optimization, and remains practical and cost-effective for deployment—a superiority that is empirically validated against multi-model baselines.
>
> ---
>
> We thank Reviewer Xgsk again for your time and for this constructive dialogue. We hope our responses, along with the planned clarifications in the revised paper, have fully addressed your concerns regarding the fairness of our comparison (W1), the novelty of our bin-packing component (W2), and our core design choice for a shared checkpoint (Q1). Given these clarifications, which highlight the 'Zero RL' challenge, the efficiency of our architecture, and our superior performance with smaller models, we respectfully hope you might reconsider your score. We truly believe our work makes a solid contribution and would be grateful for your support.

---

> ### Author Response · Authors · 2025-11-28
>
> Dear Reviewer Xgsk,
>
> We hope this message finds you well.
>
> As the discussion period is drawing to a close, we wanted to follow up to ensure you haven't missed our detailed response, in which we strove to address your specific concerns regarding the experimental setup and method design:
>
> * On Baseline Fairness (W1): We clarified that MARS operates in a "Zero RL" setting (no SFT, no distillation) using publicly curated data, comparable to baselines. We highlighted that MARS (7B) outperforms baselines like WebThinker (32B) despite the model size disadvantage, validating the fairness and strength of our comparison.
>
> * On Bin-Packing (W2): We acknowledged that while the algorithm (FFD) is standard, its application is a critical engineering enabler for parallel information processing, solving the "context bottleneck" in dual-system agents.
>
> * On Shared Checkpoint (Q1): We elaborated on how the single-model design aligns with Dual-Process Theory (one mind, two modes) and facilitates a mutually improving effect between System 1 and System 2, offering superior efficiency over separate-model approaches.
>
> We believe these clarifications directly resolve the concerns raised in your initial review.
>
> We would be deeply grateful if you could spare a moment to review our response before the rebuttal period concludes. We remain fully available to provide any further clarifications you might need.
>
> Best regards,
>
> The Authors

---

### Author Response · Authors · 2025-11-16
**General Response**

We sincerely thank all reviewers for their time, diligent work, and constructive feedback on our manuscript. Your insights are invaluable for strengthening our work.

We have carefully considered all comments. We recognize that some of the critical points raised, particularly concerning the novelty of our dual-system architecture (MARS) and our experimental choices, may benefit from a more detailed clarification.

To thoroughly address every valuable question, we have prepared detailed, point-by-point responses for each reviewer, which follow this general note.

**Given the depth of the questions and the technical details involved, our responses are necessarily comprehensive. We kindly ask for your patience in reading our detailed replies, as they contain important clarifications regarding our method's novelty, evaluation, and core contributions. We deeply appreciate the extra time and effort this may require.**

Furthermore, in direct response to your feedback, we have submitted a revised manuscript and updated supplementary materials. All significant changes made based on your suggestions have been highlighted in red text within the manuscript for your convenience.

We believe these clarifications and revisions fully address the initial concerns. We thank you once again for your time and guidance, and we respectfully ask that you re-evaluate our work based on this new information.

---

### Author Response · Authors · 2025-12-01
**Summary of Rebuttal and Key Updates for Submission #16220**

Dear Area Chair,

We thank the reviewers for their constructive feedback. We respectfully highlight that **despite our provision of comprehensive responses and new ablation experiments on Nov 16, none of the reviewers have engaged in the discussion phase or acknowledged these updates**.

As our rebuttal fundamentally resolves the stated weaknesses (particularly regarding system contributions and fairness), we kindly ask the Area Chair to review the following key outcomes which may not yet be reflected in the current ratings:

1. **Verification of Framework Effectiveness (New Experiments)**: To address Reviewer f7bk’s concern regarding the specific contribution of System 1, we conducted **a new ablation study** during the rebuttal:
    * Experiment: We disabled the co-evolved System 1, forcing System 2 to process raw tool outputs directly.
    * Result: Performance on the HLE benchmark dropped significantly from 7.38% to 5.47%.
    * Conclusion: This 1.91% drop empirically proves that System 1 is not a passive component but actively co-evolves to provide critical information distillation, directly refuting concerns about "limited conceptual novelty".

2. **Clarification on Fairness and "Zero RL" Setting**: Reviewers Xgsk and U9pi raised concerns about baseline fairness (model size/data). We clarified two critical facts:
    * **Zero RL vs. SFT**: MARS (7B/8B) is trained via RL from scratch ("Zero RL") without any Supervised Fine-Tuning (SFT) or distillation. In contrast, key baselines like WebThinker use 32B models and rely on SFT.
    * Superiority: Despite the disadvantages in model size and lack of SFT, MARS outperforms these larger, supervised baselines on HLE. This confirms the efficiency of our Multi-Agent RL framework compared to standard SFT approaches.
    * **Data Parity**: We confirmed that our training data is curated strictly from public sources (Appendix C), ensuring parity with baselines.

3. **Technical Soundness of Credit Assignment**: Addressing Reviewers iRwP and f7bk on how credit is assigned with a shared reward:
    * Mechanism: We clarified that we use **decoupled loss computation** (gradients applied to non-overlapping token sets) and **Group-Specific Advantage Estimation**. This ensures System 1 is optimized solely for summarization utility while System 2 is optimized for planning, effectively disentangling the learning signals despite the shared reward.

4. **Factual Corrections**
    * WebThinker Scores (Reviewer U9pi): We clarified to Reviewer U9pi that the discrepancy in reported scores arises because we strictly used **the Official HLE Evaluation Prompt** (full 2,154 questions), whereas the baseline paper used a custom prompt and a small subset. This ensures a fair and standardized comparison on the official benchmark.
    * Table Errors: We have corrected the formatting errors in Table 1 as pointed out by Reviewer f7bk.

Given that we have explicitly addressed the reviewers' concerns with new data and clarifications, we hope this summary assists in your final assessment.


Best regards,

The Authors

---

### Meta-Review · Area_Chair_yj22 · 2025-12-15

**Summary:**

The paper proposes a multi-agent system, with System 1 and System 2, akin to human cognition systems, for tool use and reasoning in complex tasks. Overall, the reviewers did not find the contributions sufficiently compelling -- three of four reviewers rated the contribution with a score of 2. The authors put in substantial effort in rebuttal. However, I do not think the rebuttal pointed out details that would have sufficiently changed the reviewers' initial views.

**Reviewer Concerns:**

The reviewers were mostly concerned about contributions and novelty of the paper. The authors put in substantial effort in explaining their views of the contribution and novelty, as well as answering more detailed questions. However, I did not see explanation that I think would change reviewer views by a large amount.

**Reviewer Scores:**

Some reviewers may have changed their scores but I think that it is unlikely that the change would have been substantial.

---

### Decision · Program_Chairs · 2026-01-26

Reject